# BG-HGNN: Toward Efficient Learning for Complex Heterogeneous Graphs

**Junwei Su**[*,†]                                                    *junweisu.cs@gmail.com*
*School of Artificial Intelligence and Data Science*
*University of Science and Technology of China*

**Lingjun Mao**[*]                                                   *l4mao@ucsd.edu*
*Department of Computer Science and Engineering*
*University of California San Diego*

**Zheng Da**                                                        *zhengda1936@gmail.com*
*Independent Researcher*

**Chuan Wu**[†]                                                      *cwu@cs.hku.hk*
*Department of Computer Science*
*The University of Hong Kong*

**Reviewed on OpenReview:** *https://openreview.net/forum?id=pkhDICLhy7*

## Abstract

Heterogeneous graphs—comprising diverse node and edge types connected through varied relations—are ubiquitous in real-world applications. Message-passing heterogeneous graph neural networks (HGNNs) have emerged as a powerful model class for such data. However, existing HGNNs typically allocate a separate set of learnable weights for each relation type to model relational heterogeneity. Despite their promise, these models are effective primarily on simple heterogeneous graphs with only a few relation types. In this paper, we show that this standard design inherently leads to parameter explosion (the number of learnable parameters grows rapidly with the number of relation types) and relation collapse (the model loses the ability to distinguish among different relations). These issues make existing HGNNs inefficient or impractical for complex heterogeneous graphs with many relation types. To address these challenges, we propose Blend&Grind-HGNN (BG-HGNN), a unified feature-representation framework that integrates and distills relational heterogeneity into a shared low-dimensional feature space. This design eliminates the need for relation-specific parameter sets and enables efficient, expressive learning even as the number of relations grows. Empirically, BG-HGNN achieves substantial gains over state-of-the-art HGNNs—improving parameter efficiency by up to 28.96 × and training throughput by up to 110.30 ×—while matching or surpassing their accuracy on complex heterogeneous graphs. The implementation for this paper can be found in the following link: https://github.com/mao1207/BG-HGNN.

## 1 Introduction

Heterogeneous graphs, characterised by their varied relations among diverse types of nodes and edges (Han, 2012; Sun & Han, 2013), are fundamental to modelling various data mining and machine learning problems. For example, in a bibliographic network, authors, papers, venues, and institutions form different node types, connected by relations such as writes, cites, and published-in; paper nodes may carry high-dimensional text

---

[*]: equal contribution
[†]: corresponding author

features, author nodes may be described by affiliation and research-area indicators, and venue nodes by categorical metadata. The inherent diversity in relations across such heterogeneous graphs can span distinct feature spaces of different dimensions, requiring specialised models adept at managing this heterogeneity. Heterogeneous Graph Neural Networks (HGNNs) have thus emerged as a promising deep-learning model for these graphs (Han, 2012; Sun et al., 2011a; Shao et al., 2022; Wang et al., 2021; Fu et al., 2020a; Zhang et al., 2018; 2022; Schlichtkrull et al., 2017; Zhu et al., 2019; Hong et al., 2019; Song & King, 2022). Unlike their homogeneous counterparts, graph neural networks (GNNs), HGNNs can process heterogeneous feature spaces and excel at capturing the relational information derived from different types of nodes and edges, playing a pivotal role in applications such as social networks (Liu et al., 2012; Wang et al., 2016; Hamilton et al., 2018), recommendation systems (Shi et al., 2019; 2015), biological networks (Yang et al., 2022b; Su & Wu, 2025), scene graph generation (Zhu et al., 2022; Yang et al., 2022a) and visual relationship detection (Lu et al., 2016; Xu et al., 2017; Zellers et al., 2018).

**HGNNs Categories.** Existing HGNNs can be broadly categorized into two approaches: **meta-path-based methods** (Han, 2012; Sun et al., 2011a; Shao et al., 2022; Wang et al., 2021; Fu et al., 2020a) and **message-passing methods** (Zhang et al., 2018; 2022; Schlichtkrull et al., 2017; Zhu et al., 2019; Hong et al., 2019) (see the related work section for a more detailed discussion). Meta-path-based HGNNs rely on predefined sequences of relations, called meta-paths (Han, 2012; Sun et al., 2011b), to guide the aggregation of information from various entity types within a heterogeneous graph. However, their applicability across different datasets and domains is limited due to the reliance on domain-specific meta-paths, which require substantial manual design and domain knowledge. In contrast, message-passing HGNNs, similar to the canonical framework of homogeneous GNNs, leverage local graph structures to facilitate messaging and generate node representations (Gilmer et al., 2017; Su & Wu, 2024). *This makes message-passing HGNNs less dependent on manual design and provides a more principled, data-driven, and flexible approach for heterogeneous graphs.* For these reasons, we focus on message-passing HGNNs in this paper, referring to them as HGNNs henceforth.

**Limitation of Existing HGNNs.** Despite their promising capabilities, current HGNNs face significant challenges when applied to complex heterogeneous graphs with numerous relation types. We attribute these challenges to two critical limitations: **parameter explosion** and **relation collapse**. To model heterogeneity—such as differences in node feature dimensions, semantic meanings, and edge types—most existing HGNNs assign a separate aggregation function and a distinct set of learnable parameters to each relation type. Concretely, for a relation type $r$, the model introduces its own transformation matrix $\mathbf{W}_r$ (or its own attention heads, message functions, or update functions), which independently projects messages from neighbors connected via relation $r$ into a shared latent space for subsequent aggregation (see Figure 1). This relation-specific design ensures that messages from "writes," "cites," and "published-in," for example, are processed through different parameter sets before being combined. While conceptually straightforward, this architecture has two major drawbacks:

- **Parameter Explosion.** Because each relation type requires its own parameter set, the total number of parameters grows at least linearly—and in multi-layer HGNNs, often multiplicatively—with the number of relations. In graphs with dozens or hundreds of relation types (e.g., knowledge graphs, biological interaction networks), this results in models that are memory-intensive, slow to train, and prone to overfitting.

- **Relation Collapse.** Different relation types may follow distinct semantic or statistical patterns (e.g., symmetric vs. asymmetric relations, sparse vs. dense relations, or relations with different feature distributions). However, after relation-specific transformations, these projected messages are typically summed or averaged together. When the transformed relation-specific messages lie in overlapping regions of the latent space, the model loses its ability to distinguish them—a phenomenon we refer to as relation collapse. As a result, the HGNN fails to capture fine-grained relational differences in complex graphs.

These two issues fundamentally degrade the expressiveness and representational power of existing HGNNs, limiting their practical applicability to large, complex heterogeneous graphs. Therefore, addressing parameter explosion and relation collapse is essential for building scalable, effective, and general-purpose HGNNs.

**Contribution.** This paper addresses two fundamental limitations of existing HGNNs—parameter explosion and relation collapse—which hinder their scalability and effectiveness on complex heterogeneous graphs. To overcome these challenges, we propose B̲lend&G̲rind-HGNN (BG-HGNN), a simple yet powerful HGNN framework that "blends and grinds" heterogeneous information into a unified low-rank feature space. Our main contributions are as follows:

1. **Diagnosing the limitations of existing HGNNs.** We systematically identify and analyze the dual challenges of parameter explosion and relation collapse through both empirical observations and theoretical investigation. First, we show that the number of learnable parameters in standard HGNNs grows rapidly with the number of relation types and formally derive their parameter complexity (Proposition 3.1). Second, we extend the notion of expressiveness from the graph isomorphism literature (Definition 1) to heterogeneous graphs and prove that commonly used HGNN mechanisms inherently suffer from relation collapse (Lemma B.1). These findings provide conceptual clarity on why existing HGNNs struggle with large relation spaces and offer guidance for designing more scalable alternatives.

2. **A unified and expressive HGNN framework.** Motivated by these insights, we propose BG-HGNN, an HGNN framework tailored for heterogeneous graphs with many relation types. BG-HGNN follows a unified feature–representation strategy that integrates heterogeneous information through two key principles: (i) type-aware random encoding, which maps node types and relation types into dense, approximately orthogonal vectors that preserve type-specific semantics; and (ii) low-rank interaction fusion, which jointly combines node attributes, node-type encodings, and relation-type encodings into a compact shared representation. By embedding all relations into the same low-dimensional latent space, BG-HGNN eliminates the need for relation-specific parameter sets—thereby avoiding the parameter explosion inherent in existing HGNNs—while still preserving relational distinctions through the encoded type signals. We further show that BG-HGNN is not only substantially more parameter-efficient but also strictly more expressive, in the sense of being able to distinguish a larger class of heterogeneous relational structures, than the canonical HGNN architectures in Eq. 2.1 with relation-wise sum/mean aggregation (Propositions 3.1 and 3.2).

3. **Extensive empirical validation.** We evaluate BG-HGNN across eleven benchmark datasets and five representative HGNN baselines. The results demonstrate that BG-HGNN achieves dramatic improvements in parameter efficiency (up to $28.96\times$ reduction) and training throughput (up to $110.30\times$ speedup), while matching or surpassing baseline accuracy on node classification and link prediction tasks. A series of ablation studies further confirm the importance of each design component in BG-HGNN.

## 2 Preliminary and Related Works

In this section, we present a brief overview of heterogeneous graph and heterogeneous graph neural networks. A more detailed and comprehensive discussion can be found in the supplementary material.

**Heterogeneous Graphs.** A heterogeneous graph can be defined as $\mathcal{G} = (\mathcal{V}, \mathcal{E}, \mathcal{T}_{\mathcal{V}}, \mathcal{T}_{\mathcal{E}}, \phi, \psi)$ where $\mathcal{V}$ denotes the set of nodes, $\mathcal{E} \subseteq \mathcal{V} \times \mathcal{V}$ represents the set of edges, $\mathcal{T}_{\mathcal{V}} = \{\tau_v^{(1)}, \ldots, \tau_v^{(m)}\}$ and $\mathcal{T}_{\mathcal{E}} = \{\tau_e^{(1)}, \ldots, \tau_e^{(n)}\}$ are the sets of node types and edge types respectively, $\phi : \mathcal{V} \to \mathcal{T}_{\mathcal{V}}$ is a function mapping each node to its type, and $\psi : \mathcal{E} \to \mathcal{T}_{\mathcal{E}}$ is a function mapping each edge to its type. Each node $v \in \mathcal{V}$ is associated with feature vectors $\mathbf{x}_v$. Feature vectors for nodes of different types might possess varying dimensions and distinct feature channels. For example, in a citation network, the node-type set may be $\mathcal{T}_{\mathcal{V}} = \{\text{Author}, \text{Paper}\}$ and the edge-type set may be $\mathcal{T}_{\mathcal{E}} = \{\text{Writes}, \text{Cites}\}$. Author nodes may include features such as research area, affiliated institution, or publication history, whereas paper nodes may include keyword distributions,

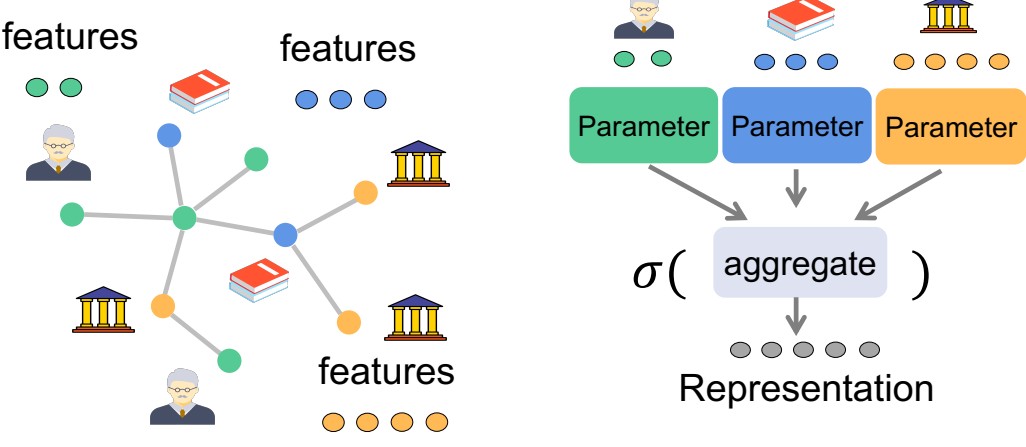

(a) Heterogeneous Graph

(b) Heterogeneous Graph Neural Network

Figure 1: Figure 1 provides an illustrative example of a heterogeneous graph in the context of a citation network. Nodes of different colors represent distinct node types (e.g., authors, papers, institutions), each associated with unique feature vectors varying in both attributes and dimensions. Figure 1(b) illustrates the structure of a single-layer heterogeneous graph neural network (HGNN). In this architecture, independent parameter spaces are employed to project heterogeneous node features into a unified latent representation. The projected embeddings are subsequently aggregated and processed through a nonlinear function $\sigma(.)$, yielding a final representation for each target node.

topic embeddings, or venue information. This diversity in both structure and feature spaces is a defining characteristic of heterogeneous graphs.

**Heterogeneous Graph Neural Networks.** HGNNs (Zhang et al., 2018; 2022; Schlichtkrull et al., 2017; Zhu et al., 2019; Hong et al., 2019; Zhou et al., 2024a;c; Zhao et al., 2021; Yang et al., 2023; 2021; 2022c) extend the foundational principles of homogeneous GNNs to address the complex and multifaceted nature of graphs composed of different types of nodes and edges. Inspired by the message-passing scheme (Schlichtkrull et al., 2017) of their homogeneous counterparts, HGNNs aggregate information from the subgraphs surrounding a target vertex to compute its embedding. This process involves collecting and integrating signals from various neighbouring nodes and edges to form a comprehensive representation of the target node. To effectively manage the heterogeneity introduced by the myriad types of relations in these graphs, HGNNs employ a distinct modelling approach. Each different relationship type is handled through a separate weight space, allowing the network to tailor its processing mechanism to the specific characteristics of each relation type. The computation in HGNNs to generate the node representation $\mathbf{h}_v^{(l)}$ at the $l$-th layer can be formulated as (the difference from GNN is highlighted in blue):

$$
\begin{aligned}
\mathbf{h}_{v,r}^{(l)} &= \text{AGGREGATE}_r^{(l)} \left( \left\{ \mathbf{h}_u^{(l-1)}, u \in \mathcal{N}_r(v) \right\} \right), \\
\mathbf{h}_v^{(l)} &= \text{COMBINE}^{(l)} \left( \mathbf{h}_v^{(l-1)}, \left\{ \mathbf{h}_{v,r}^{(l)}, r \in \mathcal{R} \right\} \right),
\end{aligned}
\tag{2.1}
$$

where $\text{AGGREGATE}_r^{(l)}$ is a relation-specific aggregation function, $\mathcal{N}_r(v)$ is the set of neighbouring nodes under relation $r$ within the neighbourhood of $v$, $\mathbf{h}_{v,r}^{(l)}$ represents the intermediate representation for relation $r$, and $\text{COMBINE}^{(l)}$ is the cross-relation combine function that blends the representations from different relations. Different choice of $\text{AGGREGATE}_r^{(l)}$ and $\text{COMBINE}^{(l)}$ results in different variants of HGNNs.

It should be noted that in the landscape of heterogeneous graph learning Han (2012); Sun et al. (2011a); Shao et al. (2022); Wang et al. (2021); Fu et al. (2020a); Qin et al. (2024); Zhong et al. (2024), there exists another branch that utilizes meta-paths Dong et al. (2017); Ferrini et al. (2024) to guide the learning process. A meta-path is a predefined sequence of relations connecting different types of entities within a heterogeneous graph,

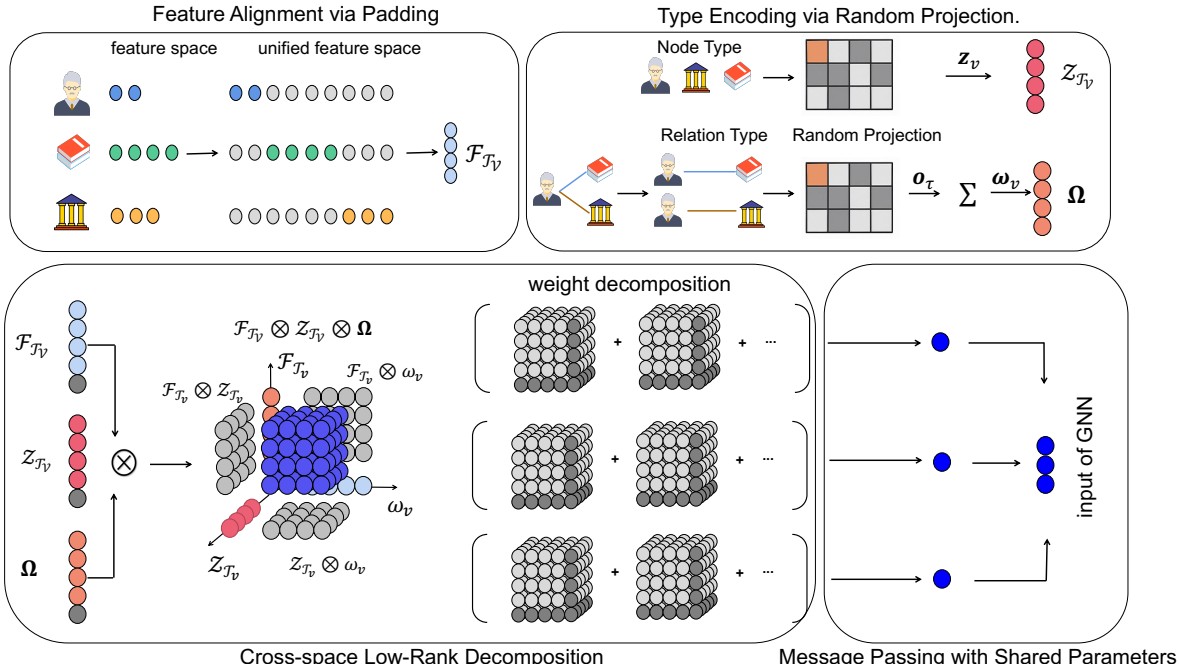

Figure 2: Overview of the BG-HGNN framework.

serving as a schema-guided pathway that reveals how different entities are interconnected through specific types of relationships. The effectiveness of meta-path-based HGNNs critically depends on the accurate selection of these meta-paths, which often demands extensive domain knowledge and considerable manual effort. The necessity for custom-designed meta-paths restricts these models' flexibility and transferability across different datasets and domains, as the relevance and availability of specific entity types and relations can dramatically vary, rendering a meta-path useful in one context but irrelevant or nonexistent in another. Given these constraints, our paper opts to explore and expand upon the message-passing HGNNs, which are more general and widely applicable.

## 3 Methodology

In this section, we introduce **Blend&Grind-HGNN (BG-HGNN)**, a concise and efficient HGNN framework tailored for complex heterogeneous graphs. We provide a formal description of its key components and theoretical analyses to demonstrate that our proposed method achieves superior parameter efficiency and expressiveness compared to existing HGNN approaches.

### 3.1 Blend&Grind-HGNN(BG-HGNN)

**Overview.** The BG-HGNN framework is composed of four key components:

1. **Unified Feature Space Construction** — aligns node features of different types into a common feature space, ensuring consistent dimensionality across all nodes.

2. **Type-aware Random Encoding** — generates dense random projections for node types and edge types to preserve type-specific semantics without introducing relation-specific parameters.

3. **Low-rank Interaction Fusion** — combines node attributes, node-type encodings, and relation-type encodings through a learnable low-rank decomposition that captures cross-modal interactions efficiently.

4. **Shared-parameter Message Passing** — applies standard homogeneous GNN layers to the fused representations, enabling structure-aware aggregation within a single shared parameter space.

Figure 2 illustrates the overall workflow, and detailed pseudocode is provided in the supplementary material.

### 3.1.1 Unified Feature Space Construction

To enable consistent processing across heterogeneous node types, BG-HGNN first projects all node features into a *shared unified feature space*. For each node type $\tau_v$, let

$$\mathcal{F}_{\tau_v} = \{f_i\}$$

denote its set of feature channels. Different node types may have distinct feature definitions (e.g., author metadata vs. paper keywords), resulting in feature vectors of varying dimensionalities. To standardize these representations, we define the *global feature space* as the union of feature channels across all node types:

$$\mathcal{F}_{\mathcal{T}_{\mathcal{V}}} = \bigcup_{\tau \in \mathcal{T}_{\mathcal{V}}} \mathcal{F}_{\tau}, \qquad D_f = |\mathcal{F}_{\mathcal{T}_{\mathcal{V}}}|.$$

Each node $v$ originally has a feature vector

$$\mathbf{x}_v \in \mathbb{R}^{|\mathcal{F}_{\tau_v}|},$$

defined only over its type-specific feature space $\mathcal{F}_{\tau_v} \subseteq \mathcal{F}_{\mathcal{T}_{\mathcal{V}}}$. To embed all nodes into the shared space, we perform a *padding-based alignment*:

$$\bar{\mathbf{x}}_v = \mathrm{PadAlign}(\mathbf{x}_v, \mathcal{F}_{\mathcal{T}_{\mathcal{V}}}) \in \mathbb{R}^{D_f},$$

where each feature channel is assigned as

$$\bar{x}_{v,i} = \begin{cases} x_{v,i}, & \text{if } f_i \in \mathcal{F}_{\tau_v}, \\ \mathrm{pad\_val}, & \text{if } f_i \notin \mathcal{F}_{\tau_v}. \end{cases}$$

Here, pad_val denotes a user-specified padding strategy used to mark feature channels that are not applicable to a given node type, such as a special token, a fixed constant, or a learnable mask. This unified representation ensures that all nodes lie in the fixed-dimensional space $\mathbb{R}^{D_f}$, enabling shared parameterization across node types and allowing the model to learn from both the presence and absence of features. If edge features are available, we apply the same alignment procedure to project them into a corresponding unified edge-feature space.

### 3.1.2 Type-aware Random Encoding.

To incorporate node-type and edge-type information into the model, we adopt a random encoding approach. This choice is motivated by two key advantages: (1) random encoding approximately preserves pairwise distances between type embeddings, thereby maintaining structural relationships without introducing bias (Wainwright, 2019); (2) unlike sparse representations (e.g., one-hot encoding), random encoding produces dense vectors that are more efficient for training and interaction modelling (Goodfellow et al., 2016).

**Node-type Encoding.** For node types, we define a random encoding function

$$\mathrm{ENC}_{\mathrm{rand}} : \mathcal{T}_{\mathcal{V}} \to \mathbb{R}^{D_{\tau_v}}$$

that maps each node type $\tau_v \in \mathcal{T}_{\mathcal{V}}$ to a dense $D_{\tau_v}$-dimensional vector:

$$\mathrm{ENC}_{\mathrm{rand}}(\tau_v) = \mathbf{z}_{\tau_v} \in \mathbb{R}^{D_{\tau_v}}, \qquad [\mathbf{z}_{\tau_v}]_j \sim \mathbb{U}(a, b) \text{ independently for } j = 1, \ldots, D_{\tau_v}.$$

Thus, each node type receives a randomly generated dense embedding whose components are independently drawn from a uniform distribution. In practice, we set

$$D_{\tau_v} = 10 \times |\mathcal{T}_{\mathcal{V}}|$$

to ensure the resulting type embeddings are approximately orthogonal. The complete set of node-type encodings is

$$\mathcal{Z}_{\mathcal{T}_{\mathcal{V}}} = \{ \mathbf{z}_\tau \mid \tau \in \mathcal{T}_{\mathcal{V}} \}.$$

**Edge-type Encoding.** Similarly, for each edge type $\tau_e \in \mathcal{T}_{\mathcal{E}}$, we generate a random dense vector $\mathbf{o}_{\tau_e} \in \mathbb{R}^{D_{\tau_e}}$ using the same procedure. To propagate relation-type information to nodes, we aggregate edge-type encodings over the local neighborhood. For each node $v$, we compute its aggregated edge-type embedding as

$$\boldsymbol{\omega}_v = \frac{1}{|\mathcal{N}(v)|} \sum_{u \in \mathcal{N}(v)} \mathbf{o}_{\psi(u,v)},$$

where $\mathcal{N}(v)$ is the set of neighbors of $v$, and $\psi(u,v) \in \mathcal{T}_{\mathcal{E}}$ denotes the type of the edge connecting $u$ to $v$. We collect all node-wise aggregated edge-type embeddings as

$$\boldsymbol{\Omega} = \{ \boldsymbol{\omega}_v \mid v \in \mathcal{V} \},$$

which later serve as complementary inputs in downstream message passing and representation learning.

### 3.1.3 Low-rank Interaction Fusion.

The final component of BG-HGNN integrates three heterogeneous sources of information for each node $v$: (i) the unified attribute vector $\bar{\mathbf{x}}_v \in \mathbb{R}^{D_f}$, (ii) the node-type encoding $\mathbf{z}_{\tau_v} \in \mathbb{R}^{D_{\tau_v}}$, and (iii) the aggregated edge-type encoding $\boldsymbol{\omega}_v \in \mathbb{R}^{D_{\tau_e}}$. To capture all cross-modal interactions among these representations, we conceptually form their Kronecker product:

$$\mathbf{H}_v = \bar{\mathbf{x}}_v \otimes \mathbf{z}_{\tau_v} \otimes \boldsymbol{\omega}_v \in \mathbb{R}^{D_f \times D_{\tau_v} \times D_{\tau_e}},$$

which enumerates every multiplicative interaction between attribute features, node-type semantics, and relation-type context. Although highly expressive, $\mathbf{H}_v$ is large in dimensionality and is therefore never materialized in practice.

**Low-rank approximation.** To obtain an efficient yet expressive representation, we leverage techniques widely used in multimodal fusion (Zadeh et al., 2017; Liu et al., 2018), and approximate the tensor interaction through a learnable low-rank approximation. Let $r$ be the rank hyperparameter, and let

$$\mathbf{w}_i^{(x)} \in \mathbb{R}^{D_h \times D_f}, \quad \mathbf{w}_i^{(z)} \in \mathbb{R}^{D_h \times D_{\tau_v}}, \quad \mathbf{w}_i^{(\omega)} \in \mathbb{R}^{D_h \times D_{\tau_e}}$$

denote learnable projection vectors for attributes, node types, and edge types, respectively. We then compute the fused representation as:

$$\mathbf{h}_v = \left( \sum_{i=1}^r \mathbf{w}_i^{(x)} \cdot \bar{\mathbf{x}}_v \right) \odot \left( \sum_{i=1}^r \mathbf{w}_i^{(z)} \cdot \mathbf{z}_{\tau_v} \right) \odot \left( \sum_{i=1}^r \mathbf{w}_i^{(\omega)} \cdot \boldsymbol{\omega}_v \right) \in \mathbb{R}^{D_h}, \tag{3.1}$$

where $\odot$ denotes element-wise multiplication and $D_h$ is the shared output dimension of each projected term. This factorized formulation corresponds to a rank-$r$ approximation of the full interaction tensor $\mathbf{H}_v$, providing the expressive power of tensor fusion. Empirically (Fig. 4(c)), a small rank ($r \approx 4-5$) is sufficient to achieve strong performance. The resulting vector $\mathbf{h}_v \in \mathbb{R}^{D_h}$ thus serves as a compact, information-rich embedding that captures heterogeneous attribute, type, and relation signals within a unified representation.

### 3.1.4 Shared-parameter Message Passing.

The fused embedding $\mathbf{h}_v$ is then used as input to standard homogeneous GNN layers. Architectures such as Graph Convolutional Networks (GCN) Kipf & Welling (2017) and Graph Attention Networks (GAT) (Veličković et al., 2018) operate directly on these unified node representations. These GNN layers propagate structural information by aggregating messages along graph edges, refining node embeddings according to the underlying topology.

Through this design, BG-HGNN seamlessly integrates feature heterogeneity, node-type semantics, relation-type context, and graph structural information into a *single shared-parameter framework.* This unification not only improves efficiency and expressiveness for heterogeneous graph learning but also enables heterogeneous and homogeneous GNNs to benefit jointly from advances in model architectures and system-level optimization.

## 3.2 Theoretical Discussion

In this subsection, we present a theoretical comparison between BG-HGNN and canonical HGNNs as defined in Eq. (2.1). Our analysis focuses on two aspects: *parameter complexity* and *expressiveness.* Parameter complexity measures the total number of learnable scalar parameters, which directly affects computational and memory efficiency, while expressiveness reflects the model's capability to distinguish between diverse relational structures and thus avoid relation collapse. The detailed proofs of the presented results can be found in the appendix.

### 3.2.1 Parameter Complexity Analysis.

To present a clean and directly comparable analysis, we make the following simplifying assumptions. We assume that all hidden representations have dimension $D$. In a canonical HGNN layer $\ell$, each relation $r \in \mathcal{R}$ is equipped with its own learnable weight matrix $\mathbf{W}_r^{(\ell)} \in \mathbb{R}^{D \times D}$, used inside the $\text{AGGREGATE}_r^{(\ell)}$ function. The $\text{COMBINE}^{(\ell)}$ function is modeled by a small MLP or linear map with $O(D^2)$ parameters, which is *shared* across all relations. Under these assumptions, the parameter complexity of canonical HGNNs and BG-HGNN scales as follows.

**Proposition 3.1.** *Consider a heterogeneous graph with $|\mathcal{R}| > 0$ relation types. Then the parameter complexity of the canonical HGNN described above is*

$$\Theta(|\mathcal{R}|LD^2),$$

*i.e., linear in both the number of relations and the number of layers. In contrast, for an BG-HGNN model with $L$ layers, the parameter complexity of its low-rank fusion module and shared-parameter message-passing layers is*

$$\Theta((r + L)D^2),$$

*where $r$ is the rank used in the low-rank decomposition. Importantly, $r$ is a small constant that does not scale with $|\mathcal{R}|$.*

Proposition 3.1 highlights that canonical HGNNs allocate a distinct parameter set for each relation at every layer. Although this dependence is linear in $|\mathcal{R}|$, the constant can be substantial in practice when $|\mathcal{R}|$ reaches the tens, hundreds, or even higher—leading to significant parameter growth and memory footprint. In contrast, BG-HGNN constructs a unified feature space and applies a low-rank fusion block (Eq. (3.1)) followed by a shared-parameter GNN. Both the unified feature dimension and the hidden dimension of the downstream GNN are fixed design choices (denoted $D$ for clarity) and do not depend on $|\mathcal{R}|$. As a result, the parameter complexity of BG-HGNN contains no $|\mathcal{R}|$ factor: all relations share the same fusion module and the same message-passing weights.

In our experiments, we find that a small constant rank (typically $r \in \{4, 5\}$) is sufficient, even when the number of relations ranges from 6 to 133. This low-rank setting allows BG-HGNN to match or surpass the performance of canonical HGNNs while using dramatically fewer parameters. Overall, these results demonstrate the practical parameter-efficiency advantage of BG-HGNN on complex heterogeneous graphs. Even

in scenarios where linear growth in relation-specific parameters may appear tolerable, reducing the number of parameters remains desirable: it mitigates memory and computation costs, enables faster convergence, and improves scalability on large-scale graphs or resource-constrained hardware.

### 3.2.2 Expressiveness Analysis.

Expressiveness, in the context of heterogeneous graph neural networks, quantifies a model's ability to distinguish between structurally different heterogeneous subgraphs by explicitly considering the variety of node and relation types.

**Definition 1** (Relative Expressiveness of HGNNs)**.** *Let $\mathscr{M}_1$ and $\mathscr{M}_2$ be two classes of HGNN models (i.e., sets of functions realized by all possible parameter settings of each architecture). We say that $\mathscr{M}_1$ is at least as expressive as $\mathscr{M}_2$ (denoted $\mathscr{M}_1 \succeq \mathscr{M}_2$) if for any pair of non-isomorphic heterogeneous subgraphs $\mathcal{G}_1, \mathcal{G}_2$,*

$$\mathscr{M}_2(\mathcal{G}_1) \neq \mathscr{M}_2(\mathcal{G}_2) \implies \mathscr{M}_1(\mathcal{G}_1) \neq \mathscr{M}_1(\mathcal{G}_2) \quad w.h.p.$$

*i.e., whenever some model in $\mathscr{M}_2$ can distinguish $\mathcal{G}_1$ and $\mathcal{G}_2$, there exists a model in $\mathscr{M}_1$ that can also distinguish them. We say that $\mathscr{M}_1$ is* strictly more expressive *than $\mathscr{M}_2$ (denoted $\mathscr{M}_1 \succ \mathscr{M}_2$) if $\mathscr{M}_1 \succeq \mathscr{M}_2$ and, in addition, there exists at least one pair of non-isomorphic heterogeneous subgraphs $\mathcal{G}_3, \mathcal{G}_4$ such that w.h.p.*

$$\mathscr{M}_1(\mathcal{G}_3) \neq \mathscr{M}_1(\mathcal{G}_4), \quad but \quad \mathscr{M}_2(\mathcal{G}_3) = \mathscr{M}_2(\mathcal{G}_4).$$

This definition captures a standard notion of expressive power for HGNNs: a model class is at least as expressive as another if it can distinguish every pair of non-isomorphic heterogeneous subgraphs that the other can. The strict version additionally requires distinguishing at least one pair that the weaker class cannot. This criterion is natural for HGNNs, whose goal is to discriminate structural and type-dependent relational patterns, and it aligns with established expressiveness analyses in the GNN literature (Sato, 2020). Thus, it provides a principled basis for comparing the representational capacity of different HGNN architectures.

**Proposition 3.2** (Relative Expressiveness)**.** *Let $\mathscr{M}_{\text{HGNN}}$ denote the class of canonical HGNNs with relation-wise mean or sum aggregation, and let $\mathscr{M}_{BG\text{-}HGNN}$ denote the class of BG-HGNN models. Suppose that the random type encodings used in BG-HGNN for node types and edge types are sampled independently from a continuous distribution in $\mathbb{R}^D$ with $D$ sufficiently large. Then we have $\mathscr{M}_{BG\text{-}HGNN} \succ \mathscr{M}_{\text{HGNN}}$.*

Proposition 3.2 states that BG-HGNN not only matches but also can exceeds the expressive power of canonical HGNNs in Eq. 2.1 with relation-wise sum/mean aggregation. The key reasons are: (i) the random type encodings preserve node-type and relation-type identities in separate channels, preventing relation collapse; and (ii) the low-rank fusion block can approximate any multilinear interaction between attributes, node types, and relation types when the rank is sufficiently large. Together, these components allow BG-HGNN to represent heterogeneous relational structures that canonical HGNNs—with relation-wise averaging and shared combination weights—fail to distinguish. This improved expressiveness is crucial for learning on complex heterogeneous graphs where relation diversity is essential for accurate prediction.

## 4 Experiments

We present an empirical evaluation to verify the effectiveness of our proposed method, BG-HGNN. Specifically, our experimental analysis is designed to address the following key questions:

- How does BG-HGNN perform compared to existing HGNN methods?

- Are the proposed design choices in each component of BG-HGNN individually effective?

Due to space constraints, we focus on presenting results directly addressing these questions in the main text. A comprehensive set of empirical results is provided in the supplementary material. The implementation of our method is available through the anonymous link provided in the abstract.

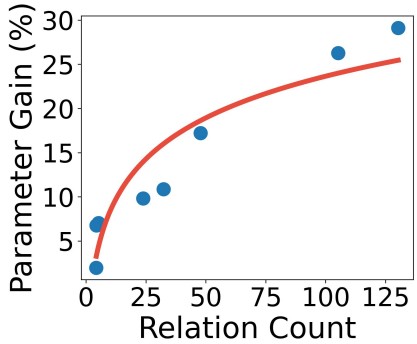
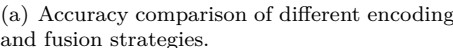

(a) Accuracy comparison of different encoding and fusion strategies.

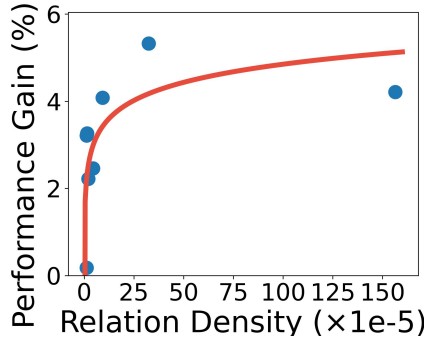

(b) Convergence Epochs required by each method to reach 95% accuracy.

Figure 3: Parameter efficiency (left) and performance gain (right) of BG-HGNN under varying relation complexity. Parameter gain reflects the parameter reduction ratio compared to RGCN (i.e., RGCN parameters divided by ours), while performance gain measures the relative accuracy improvement. Relation Count is the number of distinct relations, and Relation Density is the ratio of relations to edges in the dataset.

### 4.1 General Setup

**Datasets and tasks.** We use GAT Veličković et al. (2018) as the base GNN for our proposed method and evaluate it on two fundamental tasks—node classification and link prediction—using a total of eleven benchmark datasets. For the node classification task, we utilize the ACM Ma et al. (2021), AIFB Zhu et al. (2019), MUTAG Rossi & Ahmed (2015), BGS Zhu et al. (2019), DBLP Tang et al. (2008), AM Ristoski et al. (2016), FREEBASE Bollacker et al. (2008), and IMDB Fu et al. (2020b) datasets. For link prediction, we employ the LastFM Fu et al. (2020b), Amazon Cen et al. (2019), and YouTube Mislove et al. (2007) datasets.

**Baselines and Evaluation Metrics.** We compare BG-HGNN against several state-of-the-art (representative) message-passing HGNN methods, including Simple-HGN Lv et al. (2021), HGT Hu et al. (2020), RGCN Schlichtkrull et al. (2017), NARS Yu et al. (2020), and SlotGAT Zhou et al. (2024b). For consistent evaluation, we adopt standard metrics (e.g., accuracy and precision) following established prior works Lv et al. (2021); Yang et al. (2023), and report averaged results across five independent trials. To ensure a fair comparison, all evaluated models are configured with three layers.

### 4.2 Results.

**Effectiveness of BG-HGNN.** We first evaluate the performance of BG-HGNN on two key tasks: node classification and link prediction. For the node classification task, we follow the experimental setup outlined in Lv et al. (2021), assessing model performance using accuracy and precision. Accuracy captures the overall classification correctness, while precision measures class-specific effectiveness, ensuring a comprehensive evaluation across all classes. Additionally, we examine the number of model parameters and training throughput to quantify efficiency. For the link prediction task, we evaluate performance using ROC-AUC and Mean Reciprocal Rank (MRR), which provide a more thorough measure of model effectiveness. The results for node classification are summarized in Table 1, and link prediction results are provided in the supplementary material. *Across both tasks, BG-HGNN demonstrates robust performance, matching or surpassing competing methods in terms of accuracy, training efficiency, and parameter utilization. Notably, the relative advantage of BG-HGNN is more pronounced on datasets with a larger number of relation types*, highlighting the effectiveness of our proposed framework. At the same time, the relative accuracy gain of BG-HGNN is not uniform across all datasets. This variation is closely related to the role of relation collapse. In datasets with a small number of relation types, such as DBLP and IMDB (each with only six relations), canonical HGNNs are less likely to suffer noticeably from relation collapse, and relation-specific parameterization can remain effective.

| | param(million)↓ | throughput↑ | time(100ep, s)↓ | mem(MB)↓ | accuracy↑ | precision↑ | param(million)↓ | throughput↑ | time(100ep, s)↓ | mem(MB)↓ | accuracy↑ | precision↑ |
|---|---|---|---|---|---|---|---|---|---|---|---|---|
| Model | | | ACM Dataset | | | | | | AIFB Dataset | | | |
| HGT | 5.92 | 1.00× | 56.26 | 7448 | 88.30±1.62 | 89.34±1.48 | 16.20 | 1.00× | 48.47 | 6282 | 92.68±1.88 | 94.74±2.27 |
| RGCN | 4.43 | 13.57× | 4.18 | 1036 | 88.46±1.73 | 89.86±1.62 | 13.88 | 5.01× | 12.54 | 918 | 94.44±1.17 | 97.06±1.55 |
| simple-HGN | 1.42 | 1.60× | 35.42 | 2518 | 86.04±2.14 | 88.55±1.89 | 20.74 | 22.57× | 6.28 | 1270 | 95.16±1.55 | 98.05±1.85 |
| NARS | 1.89 | 18.87× | 3.12 | 952 | 90.73±1.41 | 90.97±1.28 | 0.82 | 35.58× | 1.41 | 548 | 83.90±2.34 | 85.96±2.17 |
| SlotGAT | 27.84 | 12.65× | 4.49 | 2794 | 90.95±1.35 | 91.23±1.17 | 1.96 | 25.73× | 1.87 | 856 | 85.99±1.96 | 87.26±1.83 |
| BG-HGNN | 0.54 | 19.98× | 2.83 | 618 | 91.88±1.04 | 92.02±0.93 | 0.81 | 36.67× | 0.90 | 417 | 97.22±0.78 | 98.44±0.62 |
| Model | | | MUTAG Dataset | | | | | | BGS Dataset | | | |
| HGT | 9.23 | 1.00× | 29.68 | 8812 | 72.06±2.27 | 70.96±2.66 | 35.11 | 1.00× | 90.59 | 25042 | 79.31±2.30 | 85.34±2.65 |
| RGCN | 6.64 | 13.57× | 7.64 | 1183 | 77.97±1.90 | 74.88±2.30 | 16.14 | 5.01× | 17.83 | 3118 | 89.91±1.19 | 91.30±1.53 |
| simple-HGN | 24.31 | 12.76× | 23.18 | 2978 | 77.45±2.31 | 85.16±1.94 | 20.74 | 6.20× | 39.90 | 12531 | 86.44±2.27 | 86.77±2.69 |
| NARS | 0.71 | 8.26× | 3.73 | 738 | 74.43±2.18 | 76.90±2.43 | 6.39 | 4.48× | 20.24 | 3247 | 73.82±2.76 | 75.43±2.54 |
| SlotGAT | 1.38 | 12.23× | 2.47 | 1465 | 75.61±2.05 | 78.44±1.97 | - | - | - | - | - | - |
| BG-HGNN | 0.71 | 110.30× | 1.74 | 1153 | 83.60±1.53 | 85.71±1.14 | 2.13 | 11.43× | 17.48 | 3078 | 94.49±0.64 | 95.24±0.62 |
| Model | | | FREEBASE Dataset | | | | | | DBLP Dataset | | | |
| HGT | - | - | - | - | - | - | 4.13 | 1.00× | 51.83 | 5924 | 94.95±0.43 | 93.93±0.52 |
| RGCN | 6.42 | 1.00× | 58.41 | 5712 | 60.73±0.84 | 67.62±0.97 | 9.17 | 1.50× | 34.55 | 878 | 93.04±0.67 | 91.96±0.74 |
| simple-HGN | 12.87 | 2.24× | 26.07 | 22584 | 61.84±1.12 | 61.95±1.34 | 20.71 | 1.27× | 40.81 | 1986 | 94.97±0.38 | 93.93±0.45 |
| NARS | - | - | - | - | - | - | 2.01 | 1.38× | 37.56 | 716 | 93.54±0.59 | 93.72±0.63 |
| SlotGAT | - | - | - | - | - | - | 1.58 | 1.05× | 49.36 | 1568 | 93.28±0.71 | 93.34±0.68 |
| BG-HGNN | 0.68 | 3.02× | 19.34 | 5348 | 64.40±0.76 | 68.83±0.88 | 1.46 | 1.39× | 37.29 | 647 | 94.79±0.46 | 94.06±0.41 |
| Model | | | AM Dataset | | | | | | IMDB Dataset | | | |
| HGT | - | - | - | - | - | - | 7.35 | 1.00× | 38.12 | 4561 | 52.57±1.84 | 48.05±2.13 |
| RGCN | 78.52 | 1.00× | 817.82 | 74128 | 89.90±0.95 | 87.58±1.07 | 5.80 | 3.61× | 10.54 | 847 | 50.18±2.07 | 50.48±1.96 |
| simple-HGN | - | - | 282.20 | 71948 | - | - | 2.93 | 1.17× | 32.68 | 1318 | 51.40±1.73 | 51.29±1.85 |
| NARS | - | - | - | - | - | - | 5.13 | 2.26× | 16.51 | 762 | 54.56±1.62 | 50.15±1.91 |
| SlotGAT | - | - | - | - | - | - | 4.35 | 1.08× | 35.31 | 1124 | 55.43±1.54 | 56.82±1.47 |
| BG-HGNN | 2.71 | 6.27× | 130.20 | 70657 | 90.25±0.82 | 88.24±0.89 | 2.91 | 3.82× | 9.88 | 682 | 53.29±1.78 | 61.98±1.23 |

Table 1: Comparative performance analysis on various datasets for the node classification task. We additionally report the wall-clock training time measured per 100 epochs and the peak GPU memory usage. Accuracy and precision are reported as mean±std when standard deviations are available; otherwise only the mean is shown. Entries marked as "-" indicate unavailable results, while "OOM" indicates that the experiment exceeded the available memory capacity on our test platform. Throughput is evaluated as the time required to complete one training epoch, with the relative improvement reported compared to the slowest baseline. The best-performing method is highlighted in green, and the second-best is highlighted in light green.

As a result, these baselines may achieve comparable or slightly better accuracy. In contrast, on datasets with many relation types, where relation-specific messages are more likely to become indistinguishable after aggregation, BG-HGNN demonstrates clearer advantages. This trend further supports our central claim that BG-HGNN is particularly well-suited for complex heterogeneous graphs with rich relational structure.

**Ability to Handle Complex Heterogeneous Graphs.** To further highlight the capability of BG-HGNN in processing complex heterogeneous graphs, we analyze its relative performance improvement compared to the representative HGCN as the number of relation types increases. The results, presented in Figure 3, demonstrate that BG-HGNN achieves significantly greater improvements in parameter efficiency and computational throughput as the relation count grows. These findings not only reaffirm the presence of parameter explosion and relation collapse issues in canonical HGNN frameworks but also underscore the effectiveness and adaptability of BG-HGNN in addressing these challenges within complex heterogeneous graph scenarios.

## 4.3 Ablation Study

**Impact of Feature Alignment and Random Encoding.** We first evaluate the effectiveness of our proposed random encoding and Kronecker product-based feature fusion methods. We compare these strategies against commonly used methods, namely, direct concatenation for feature fusion and one-hot encoding for type encoding. The experiments focus on accuracy as a measure of model performance and the convergence rate as an indicator of learning efficiency. Figure 4(a) presents the accuracy comparison across various encoding and fusion combinations. The results clearly illustrate that the Kronecker product consistently achieves higher accuracy than direct concatenation across all datasets. This confirms that explicitly modeling interactions among heterogeneous features through the Kronecker product effectively captures nuanced relational information within heterogeneous graphs. We further analyze convergence efficiency by measuring the number of epochs required to achieve a target accuracy of 95 %. The results, summarized in Figure 4(b), indicate that although one-hot encoding combined with the Kronecker product achieves accuracy levels similar to random encoding, it suffers from notably slower convergence. In contrast, using random projection encoding significantly enhances training efficiency, confirming our method's advantage in terms of both accuracy and computational throughput. In addition, we present a robustness study on the randomness and dimension-

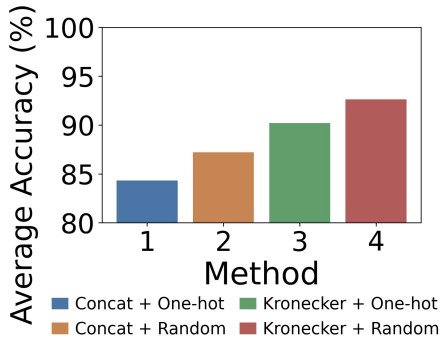
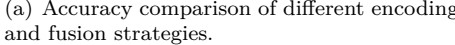
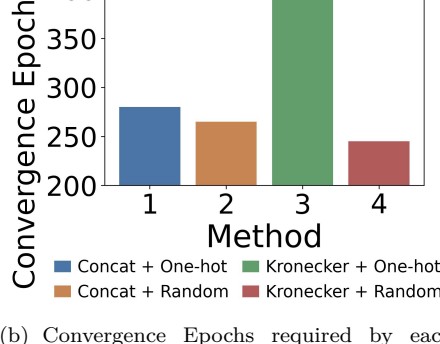

(a) Accuracy comparison of different encoding and fusion strategies.

(b) Convergence Epochs required by each method to reach 95% accuracy.

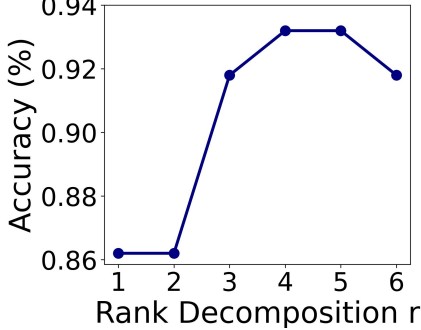
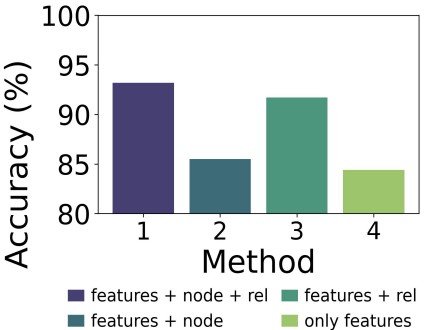

(c) Accuracy with different values of low-rank hyperparameter $r$.

(d) Accuracy comparison of BG-HGNN using various feature sets.

Figure 4: Comparison across encoding/fusion methods and modeling strategies. The first two figures show accuracy and convergence efficiency, where C and K denote Concat and Kronecker fusion, and O and R represent One-hot and Random relation encoding. Each combination reflects a distinct design choice. The latter two show the impact of rank selection and feature combinations on BG-HGNN. In figure (d), F, N, and R denote input node features, node type, and relation type, and each method corresponds to a different combination of these features.

ality of the encoding in Table 6 and Fig. 12. The results show that the random encoding is robust to the sampling process and, as expected, becomes largely insensitive to the encoding dimension once it exceeds a modest threshold relative to the number of relations in the dataset.

**Effect of the Low-rank Decomposition Parameter.** We investigate the sensitivity of model performance to the rank hyperparameter $r$ used in the low-rank decomposition step of BG-HGNN. The corresponding results are shown in Fig.4(c). The experiments demonstrate that performance initially improves with increasing $r$, subsequently reaching a plateau. Beyond this point, further increments in $r$ lead to minimal performance gain and eventually slight performance degradation due to increased computational complexity. Empirical evaluations indicate that choosing $r = 4/5$ provides optimal balance between accuracy and computational efficiency, aligning with findings from prior literature on low-rank adaptation methodsLiu et al. (2018). Hence, we recommend $r = 4/5$ as suitable practical choices for deploying BG-HGNN.

**Benefit of Encoding Node and Relation Types.** To further validate our approach of encoding node types and relation types in addition to using the original node features, we conducted experiments comparing several configurations. Specifically, we assessed the model accuracy under four distinct feature setups: (1) original node features alone; (2) original features combined with node type encoding; (3) original features combined with relation encoding; and (4) original features combined with both node and relation type

encodings. Results, shown in Fig. 11, indicate that integrating both node and relation encodings achieves the highest accuracy consistently across the evaluated datasets. This outcome underscores the importance and effectiveness of simultaneously preserving multiple sources of heterogeneous information within a unified feature representation, thereby significantly enhancing model performance.

## 5  Concluding Discussion

**Conclusion.**  We introduced BG-HGNN, a new HGNN framework designed to overcome two longstanding limitations of heterogeneous graph neural networks: parameter explosion and relation collapse. By unifying heterogeneous information into a shared representation space and employing a low-rank interaction fusion mechanism, BG-HGNN provides a parameter-efficient and highly expressive alternative to canonical HGNNs. Our theoretical analysis establishes both its improved parameter scaling and superior expressive power, while extensive experiments confirm its strong empirical performance and scalability on complex heterogeneous graphs.

**Limitations and Future Work.**  A primary limitation of the present work is its focus on static heterogeneous graphs. In many real-world applications, node features, edge types, and relational structures evolve over time (Su et al., 2025; 2023; Su et al.). Adapting BG-HGNN to dynamic or streaming heterogeneous graphs—where the model must update efficiently in response to continual graph changes—remains an important direction for future research. Further opportunities include developing incremental or continual learning variants of BG-HGNN, exploring temporal extensions of the type encoding and fusion mechanism, and integrating the framework with forecasting or anomaly detection tasks in dynamic relational environments.

**Acknowledgement.**  We would like to thank the anonymous reviewers and editors for their helpful comments. This work was supported in part by grants from Hong Kong RGC under the contracts 17203522 (GRF), C7004-22G (CRF), C5032-23G (CRF), and T43-513/23-N (TRS).

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

## A Broader Impact

This work is primarily methodological and aims to improve the efficiency and scalability of heterogeneous graph neural networks. The proposed framework does not introduce new application-specific risks beyond those already present in graph-based machine learning systems. However, when applied to sensitive relational datasets—such as social networks, financial transaction graphs, or biomedical interaction networks—standard concerns related to privacy, fairness, and potential misuse may arise. Practitioners applying the proposed methods should therefore follow established best practices for responsible machine learning, including careful dataset governance, privacy protection, and fairness evaluation.

## B Theoretical Proofs

### B.1 Proof of Parameter Complexity Results

In this section, we provide rigorous proofs for Proposition 3.1 which establish the parameter complexity of canonical HGNNs and our proposed BG-HGNN model, respectively.

*Proof.* We analyze the two architectures separately.

**Canonical HGNN.** In each layer $\ell$, canonical HGNNs allocate a separate aggregation weight matrix for each relation $r \in \mathcal{R}$. Each such weight matrix has size $D \times D$, contributing $D^2$ parameters. Thus, the AGGREGATE portion of a single HGNN layer contributes:

$$|\mathcal{R}| \cdot D^2 \quad \text{parameters.}$$

The COMBINE function is shared across relations and adds an additional $O(D^2)$ parameter, which does not depend on $|\mathcal{R}|$ and is dominated by the AGGREGATE term when $|\mathcal{R}| > 1$. Therefore, the total parameter count of a single HGNN layer is

$$\Theta(|\mathcal{R}|D^2).$$

Stacking $L$ such layers yields total complexity

$$\Theta(|\mathcal{R}|LD^2).$$

**BG-HGNN.** BG-HGNN eliminates relation-specific parameter matrices entirely. The parameters come from two sources:

1. **Low-rank fusion block.** The fusion block (Eq. (3.1)) uses $r$ rank components, each consisting of three projection vectors of dimension $D$. Thus, the fusion block has:

$$3rD \quad \text{learnable parameters.}$$

   The output of the fusion block is projected into a $D$-dimensional node representation via a learnable vector or matrix, contributing another $O(D^2)$ parameters. Since $r$ is a constant and $D$ is typically much larger, the overall complexity of the fusion module scales as:

$$\Theta(rD^2).$$

2. **Shared-parameter GNN layers.** After the fusion block, BG-HGNN performs message passing using a standard homogeneous GNN with $L$ layers. Each such layer has a shared weight matrix in $\mathbb{R}^{D \times D}$, giving:

$$LD^2 \quad \text{parameters.}$$

Summing the two components yields:

$$\Theta(rD^2) + \Theta(LD^2) = \Theta((r + L)D^2).$$

Crucially, this expression contains *no dependence on* $|\mathcal{R}|$, because all relations share both the fusion module and the downstream GNN weights. □

### B.2 Proof of Expressiveness

**Lemma B.1** (Relation Collapse). *Consider three relation types $r_1, r_2, r_3$, and suppose that for each edge of type $r_i$ the (scalar) feature of the neighbor is drawn independently from a normal distribution $\mathcal{N}(\mu_i, 1)$ with $\mu_1 = 0$, $\mu_2 = -1$, and $\mu_3 = 1$. Construct two heterogeneous ego-subgraphs around a fixed center node:*

- *$\mathcal{G}_1$: the center node is connected only via relation $r_1$;*

- *$\mathcal{G}_2$: the center node is connected via $r_1$, $r_2$, and $r_3$, with the same number of neighbors per relation.*

*Consider the class $\mathscr{M}_{\mathrm{HGNN}}$ of canonical HGNNs whose use relation-wise mean or sum for aggregation. Then:*

1. *The expected representation of the center node produced by any model in $\mathscr{M}_{\mathrm{HGNN}}$ is the same for $\mathcal{G}_1$ and $\mathcal{G}_2$, so no such HGNN can distinguish the two ego-subgraphs in expectation.*

2. *In contrast, with probability $1$ over the random type encodings, there exists a parameter setting of BG-HGNN for which the expected representations of the center node for $\mathcal{G}_1$ and $\mathcal{G}_2$ differ.*

*Proof.* We first analyze the canonical HGNN.

**(1) Canonical HGNN cannot distinguish $\mathcal{G}_1$ and $\mathcal{G}_2$ in expectation.** Fix a center node $v$. For a relation type $r$, let

$$m_r(v) = \frac{1}{|\mathcal{N}_r(v)|} \sum_{u \in \mathcal{N}_r(v)} x_u$$

denote the mean of neighbor features under relation $r$, where $\mathcal{N}_r(v)$ is the multiset of neighbors of $v$ connected via relation $r$, and $x_u$ is the (scalar) neighbor feature.

By assumption, in $\mathcal{G}_1$ all neighbors are connected via $r_1$ and $x_u \sim \mathcal{N}(0, 1)$, so by linearity of expectation,

$$\mathbb{E}[m_{r_1}(v) \mid \mathcal{G}_1] = \mu_1 = 0, \quad \mathbb{E}[m_{r_2}(v) \mid \mathcal{G}_1] = 0, \quad \mathbb{E}[m_{r_3}(v) \mid \mathcal{G}_1] = 0,$$

where we take the mean for relations with no neighbors (here $r_2$ and $r_3$ in $\mathcal{G}_1$) to be $0$ for definiteness.

In $\mathcal{G}_2$, the center node has the same number of neighbors under each relation type $r_1, r_2, r_3$, with

$$x_u \sim \begin{cases} \mathcal{N}(0, 1), & \text{if } (u, v) \text{ has type } r_1, \\ \mathcal{N}(-1, 1), & \text{if type } r_2, \\ \mathcal{N}(1, 1), & \text{if type } r_3. \end{cases}$$

Hence,

$$\mathbb{E}[m_{r_1}(v) \mid \mathcal{G}_2] = 0, \quad \mathbb{E}[m_{r_2}(v) \mid \mathcal{G}_2] = -1, \quad \mathbb{E}[m_{r_3}(v) \mid \mathcal{G}_2] = 1.$$

A canonical HGNN in $\mathscr{M}_{\mathrm{HGNN}}$ computes, at the center node,

$$h_v = \sigma\left( \sum_{r \in \{r_1, r_2, r_3\}} m_r(v) \right),$$

where $\sigma$ is a fixed pointwise nonlinearity (possibly preceded or followed by a shared linear map that does not depend on $r$). Thus

$$\sum_r m_r(v) = \begin{cases} m_{r_1}(v) + 0 + 0, & \text{in } \mathcal{G}_1, \\ m_{r_1}(v) + m_{r_2}(v) + m_{r_3}(v), & \text{in } \mathcal{G}_2. \end{cases}$$

Taking expectations and using the calculations above,

$$\mathbb{E}\left[ \sum_r m_r(v) \mid \mathcal{G}_1 \right] = 0, \quad \mathbb{E}\left[ \sum_r m_r(v) \mid \mathcal{G}_2 \right] = 0 + (-1) + 1 = 0.$$

Since the subsequent operations are shared and deterministic given $\sum_r m_r(v)$, it follows that

$$\mathbb{E}[h_v \mid \mathcal{G}_1] = \mathbb{E}[h_v \mid \mathcal{G}_2].$$

Therefore, no model in $\mathscr{M}_{\mathrm{HGNN}}$ can distinguish $\mathcal{G}_1$ and $\mathcal{G}_2$ in expectation.

**(2) BG-HGNN can distinguish $\mathcal{G}_1$ and $\mathcal{G}_2$ with appropriate parameters.** In BG-HGNN, each edge type $\tau_e \in \mathcal{T}_{\mathcal{E}}$ is assigned a dense random encoding $\mathbf{o}_{\tau_e} \in \mathbb{R}^{D_t}$, sampled independently from a continuous distribution. For a node $v$, the aggregated edge-type encoding is

$$\boldsymbol{\omega}_v = \frac{1}{|\mathcal{N}(v)|} \sum_{u \in \mathcal{N}(v)} \mathbf{o}_{\psi(u,v)}.$$

In $\mathcal{G}_1$, all neighbors are of type $r_1$, so
$$\boldsymbol{\omega}_v^{(1)} = \mathbf{o}_{r_1}.$$
In $\mathcal{G}_2$, the neighbors are evenly split among $r_1, r_2, r_3$, so

$$\boldsymbol{\omega}_v^{(2)} = \tfrac{1}{3}(\mathbf{o}_{r_1} + \mathbf{o}_{r_2} + \mathbf{o}_{r_3}).$$

Since the encodings $\mathbf{o}_{r_1}, \mathbf{o}_{r_2}, \mathbf{o}_{r_3}$ are sampled independently from a continuous distribution, we have

$$\mathbb{P}\big[\mathbf{o}_{r_1} = \tfrac{1}{3}(\mathbf{o}_{r_1} + \mathbf{o}_{r_2} + \mathbf{o}_{r_3})\big] = 0,$$

i.e., with probability 1 we have $\boldsymbol{\omega}_v^{(1)} \neq \boldsymbol{\omega}_v^{(2)}$.

Now consider an instance of BG-HGNN that ignores node attributes and node-type encodings in the fusion block and focuses only on $\boldsymbol{\omega}_v$ (this can be implemented, e.g., by setting the corresponding projection weights to zero). Because $\boldsymbol{\omega}_v$ takes different values on $\mathcal{G}_1$ and $\mathcal{G}_2$, and the fusion block plus downstream GNN is at least as expressive as a linear map followed by a nonlinearity, we can choose parameters so that the resulting node embeddings satisfy
$$\mathbb{E}[\mathbf{h}_v \mid \mathcal{G}_1] \neq \mathbb{E}[\mathbf{h}_v \mid \mathcal{G}_2].$$

For example, any linear functional that separates $\boldsymbol{\omega}_v^{(1)}$ and $\boldsymbol{\omega}_v^{(2)}$ followed by a strictly monotone nonlinearity will suffice. Thus, there exists a parameter setting of BG-HGNN that distinguishes $\mathcal{G}_1$ and $\mathcal{G}_2$ in expectation. This completes the proof. $\qquad\square$

Lemma B.1 illustrates a concrete form of *relation collapse*: when the cross-relation aggregation in a canonical HGNN depends only on aggregated means across all relations, symmetric or overlapping distributions over relations can lead to indistinguishable representations, even though the underlying relation patterns differ. In contrast, BG-HGNN preserves relation-type information in separate channels through type encodings and the fusion block, avoiding this collapse.

We now prove Proposition 3.2.

*Proof of Proposition 3.2.* We prove the two parts separately.

**(i) $\mathscr{M}_{\mathbf{BG\text{-}HGNN}} \succeq \mathscr{M}_{\mathrm{HGNN}}$.** Fix any canonical HGNN architecture in $\mathscr{M}_{\mathrm{HGNN}}$ with $L$ layers and hidden dimension $D$. Suppose there exist two heterogeneous subgraphs $\mathcal{G}_1$ and $\mathcal{G}_2$ such that the HGNN distinguishes them, i.e.,
$$f_{\mathrm{HGNN}}(\mathcal{G}_1) \neq f_{\mathrm{HGNN}}(\mathcal{G}_2)$$
for some parameter setting. We show that BG-HGNN can also distinguish these subgraphs with high probability.

*Step 1: Any HGNN-distinguishable difference must appear in the heterogeneous input.* The computation in a canonical HGNN layer depends only on (i) node attributes $\mathbf{X}$, (ii) node types $\mathcal{T}_{\mathcal{V}}$, (iii) edge types $\mathcal{T}_{\mathcal{E}}$, (iv)

the type-assignment maps $\phi, \psi$, and (v) the adjacency structure $\mathcal{E}$. Since $f_{\text{HGNN}}(\mathcal{G}_1) \neq f_{\text{HGNN}}(\mathcal{G}_2)$, the two subgraphs must differ in at least one of these components:

$$(\mathcal{V}, \mathcal{E}, \mathcal{T}_{\mathcal{V}}, \mathcal{T}_{\mathcal{E}}, \phi, \psi, \mathbf{X}).$$

Thus, there exists at least one node $v$ for which the heterogeneous "signature" differs between $\mathcal{G}_1$ and $\mathcal{G}_2$.

*Step 2: BG-HGNN preserves heterogeneous differences injectively w.h.p.* In BG-HGNN, each node type $\tau \in \mathcal{T}_{\mathcal{V}}$ and each edge type $\tau_e \in \mathcal{T}_{\mathcal{E}}$ is assigned a random encoding vector sampled i.i.d. from a continuous distribution in $\mathbb{R}^{D_t}$. With probability 1, all such vectors are distinct and linearly independent.

For any node $v$, the type-aware encoding triplet

$$(\bar{\mathbf{x}}_v, \ \mathbf{z}_{\tau_v}, \ \boldsymbol{\omega}_v)$$

is formed from: (i) padded raw features $\bar{\mathbf{x}}_v$, (ii) the node-type vector $\mathbf{z}_{\tau_v}$, and (iii) the aggregated edge-type encoding

$$\boldsymbol{\omega}_v = \frac{1}{|\mathcal{N}(v)|} \sum_{u \in \mathcal{N}(v)} \mathbf{o}_{\psi(u,v)}.$$

Since $\mathbf{z}_{\tau_v}$ and $\mathbf{o}_{\psi(u,v)}$ differ whenever the node type or edge type differs, and since $\boldsymbol{\omega}_v$ is an affine function of the $\mathbf{o}$'s, any difference in $(\mathcal{T}_{\mathcal{V}}, \mathcal{T}_{\mathcal{E}}, \phi, \psi)$ or $\mathbf{X}$ leads to a difference in at least one coordinate of $(\bar{\mathbf{x}}_v, \mathbf{z}_{\tau_v}, \boldsymbol{\omega}_v)$ for at least one node.

Thus, for any heterogeneous difference between $\mathcal{G}_1$ and $\mathcal{G}_2$, BG-HGNN's encoded representation also differs with probability 1.

*Step 3: The low-rank fusion block can preserve and propagate these differences.* For each node $v$, the fusion block computes:

$$\mathbf{h}_v = \sum_{i=1}^{r} (\mathbf{w}_i^{(x)} \cdot \bar{\mathbf{x}}_v) \, (\mathbf{w}_i^{(z)} \cdot \mathbf{z}_{\tau_v}) \, (\mathbf{w}_i^{(\omega)} \cdot \boldsymbol{\omega}_v).$$

Let

$$(\bar{\mathbf{x}}_v^{(1)}, \mathbf{z}_{\tau_v}^{(1)}, \boldsymbol{\omega}_v^{(1)}), \qquad (\bar{\mathbf{x}}_v^{(2)}, \mathbf{z}_{\tau_v}^{(2)}, \boldsymbol{\omega}_v^{(2)})$$

denote the encoded features of a node $v$ in $\mathcal{G}_1$ and $\mathcal{G}_2$, respectively. From Step 2, these differ for at least one coordinate; assume for concreteness that

$$\bar{\mathbf{x}}_v^{(1)}[j] \neq \bar{\mathbf{x}}_v^{(2)}[j]$$

for some coordinate $j$ (the argument is identical if the difference arises in $\mathbf{z}_{\tau_v}$ or $\boldsymbol{\omega}_v$).

Choose the first set of projection vectors in the rank-$r$ fusion block as:

$$\mathbf{w}_1^{(x)} = e_j, \qquad \mathbf{w}_1^{(z)} = \mathbf{1}, \qquad \mathbf{w}_1^{(\omega)} = \mathbf{1},$$

where $e_j$ is the $j$-th standard basis vector and $\mathbf{1}$ is the all-ones vector. Then:

$$(\mathbf{w}_1^{(x)} \cdot \bar{\mathbf{x}}_v^{(1)}) = \bar{\mathbf{x}}_v^{(1)}[j] \neq \bar{\mathbf{x}}_v^{(2)}[j] = (\mathbf{w}_1^{(x)} \cdot \bar{\mathbf{x}}_v^{(2)}).$$

Since the other two inner products are positive scalars (and can be made equal or fixed by construction), the first rank component satisfies:

$$(\mathbf{w}_1^{(x)} \cdot \bar{\mathbf{x}}_v^{(1)})(\mathbf{w}_1^{(z)} \cdot \mathbf{z}_{\tau_v}^{(1)})(\mathbf{w}_1^{(\omega)} \cdot \boldsymbol{\omega}_v^{(1)}) \ \neq \ (\mathbf{w}_1^{(x)} \cdot \bar{\mathbf{x}}_v^{(2)})(\mathbf{w}_1^{(z)} \cdot \mathbf{z}_{\tau_v}^{(2)})(\mathbf{w}_1^{(\omega)} \cdot \boldsymbol{\omega}_v^{(2)}).$$

Hence, even with $r = 1$, the fusion block produces distinct outputs for $\mathcal{G}_1$ and $\mathcal{G}_2$ for this node $v$. For larger $r$, the expressive capacity only increases. Thus, BG-HGNN preserves (and can amplify) any heterogeneous difference established in Step 2.

*Conclusion of (i).* Since every heterogeneous difference detectable by a canonical HGNN leads to a difference in BG-HGNN's encoded features (Step 2), and since the fusion block can propagate this difference to the output (Step 3), BG-HGNN distinguishes any pair that a canonical HGNN can. Therefore:

$$\mathscr{M}_{\text{BG-HGNN}} \succeq \mathscr{M}_{\text{HGNN}}.$$

**(ii) Strictness:** $\mathscr{M}_{\mathbf{BG\text{-}HGNN}} \succ \mathscr{M}_{\mathrm{HGNN}}$. Lemma B.1 constructs heterogeneous ego-subgraphs $(\mathcal{G}_1, \mathcal{G}_2)$ such that:

- A canonical HGNN collapses their relational differences due to mean/sum aggregation and produces identical expected representations.

- BG-HGNN, using random and independent type encodings, assigns distinct relation-type vectors to $r_1, r_2, r_3$. These propagate into $\boldsymbol{\omega}_v$, which differ for $\mathcal{G}_1$ and $\mathcal{G}_2$ with probability 1. The fusion block then maps these into different node embeddings for some parameter setting.

Thus, there exists *at least one* pair of subgraphs distinguished by BG-HGNN but collapsed by all canonical HGNNs, establishing strictness.

Combining (i) and (ii), we conclude:
$$\mathscr{M}_{\mathrm{BG\text{-}HGNN}} \succ \mathscr{M}_{\mathrm{HGNN}},$$
i.e., BG-HGNN is strictly more expressive than canonical HGNN architectures. $\qquad\square$

In summary, BG-HGNN retains (and in fact strictly extends) the expressive power of canonical HGNNs. This design prevents relation collapse and enables finer-grained discrimination of heterogeneous relational structures.

## C  Further Discussion

### C.1  Illustration of Relation Collapse

**Definition 2** (Relation Collapse)**.** *Let $\mathcal{M}$ denote a class of heterogeneous graph neural networks (HGNNs) that compute node representations via relation-specific aggregation followed by a relation-agnostic combination operator, such as summation or averaging. We say that $\mathcal{M}$ suffers from* relation collapse *if there exist two heterogeneous ego-subgraphs $G_1$ and $G_2$ with different relation compositions such that*

$$h_{\mathcal{M}}(G_1) = h_{\mathcal{M}}(G_2),$$

*even though the relational structures in $G_1$ and $G_2$ are different. In this case, the model fails to distinguish distinct relation patterns after aggregation and therefore loses the ability to preserve heterogeneous relational semantics.*

To provide an intuitive understanding of the *relation collapse* phenomenon, we present a conceptual illustration in Fig. 5. In canonical message-passing HGNNs, information from neighbors under different relation types is first processed through relation-specific transformations. The resulting relation-conditioned messages are then merged through a shared cross-relation aggregation operator, such as mean or sum.

Although this design projects heterogeneous information into a unified latent space, the subsequent combination step is relation-agnostic. As a consequence, distinct relational patterns may yield identical aggregated representations when their transformed messages overlap in the latent space. In such cases, the model becomes insensitive to the underlying relation composition, even though the heterogeneous structures themselves are different.

Figure 5 illustrates this phenomenon. Two heterogeneous ego-graphs with different relation configurations produce relation-specific intermediate messages that, after transformation, occupy overlapping regions in the representation space. Once these messages are combined using a relation-agnostic operator, the resulting node embeddings become indistinguishable. Consequently, the HGNN can no longer preserve the semantic differences among relations, which leads to *relation collapse*.

In contrast, BG-HGNN preserves relation distinctions by injecting node-type and relation-type information directly into the node representation before message passing. The fused representation retains heterogeneous relational signals within dedicated feature channels, enabling the subsequent shared-parameter GNN to

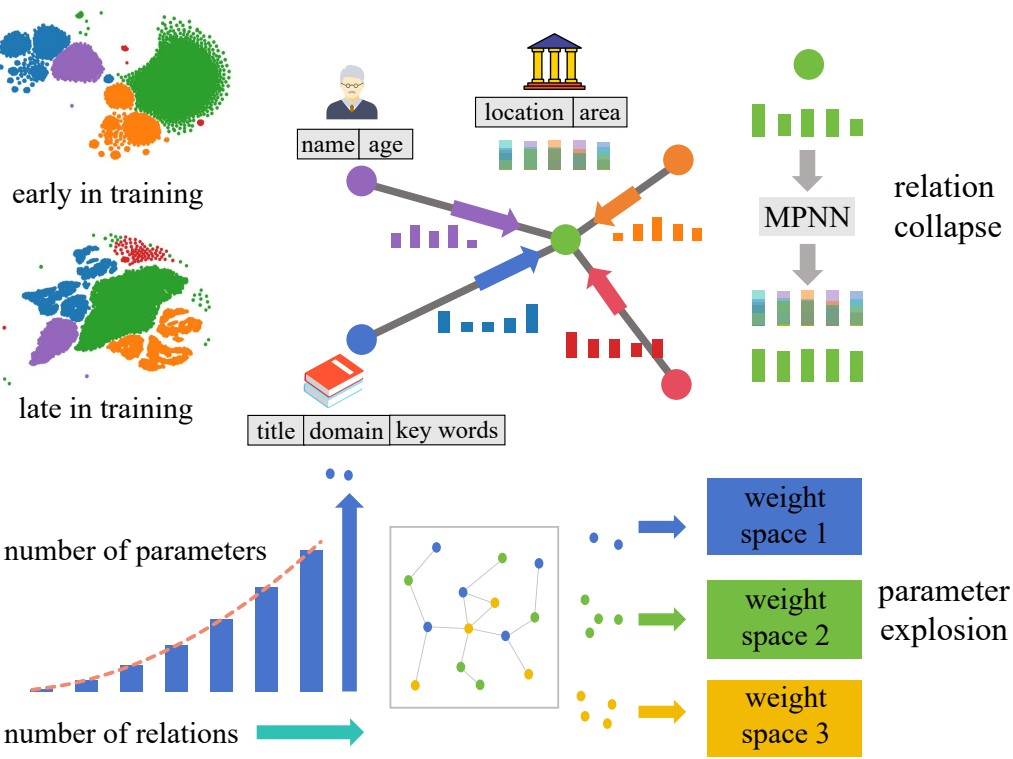

Figure 5: Illustration of relation collapse in canonical HGNNs. Relation-specific messages are first transformed independently and then combined through a relation-agnostic aggregation operator (e.g., mean or sum). When the transformed messages overlap in the latent space, the final representation becomes insensitive to the underlying relation composition, causing distinct heterogeneous structures to yield identical embeddings.

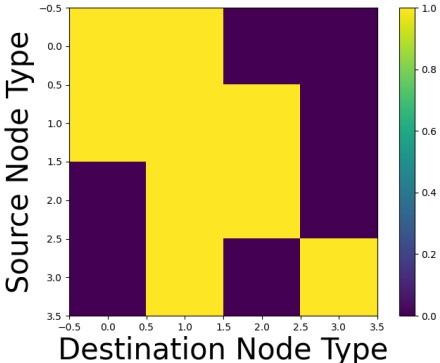 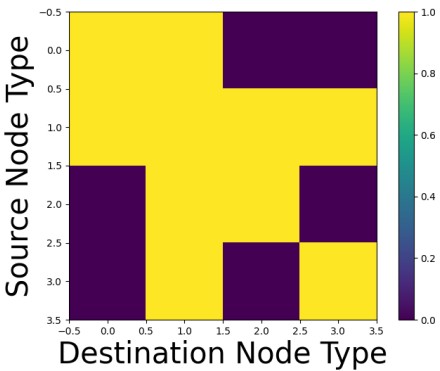

(a) Matrix of binarized attention/activation scores after training. The threshold for binarization is 0.1.

(b) Binary matrix of ground truth representing the interaction of different node-defined meta-paths in the ACM dataset.

Figure 6: The plots depict the inferred attention/activation from our method and the ground truth for meta-path discovery featuring four node types: Author, Paper, Term, and Subject. Both matrices are of size $50 \times 50$, showing the interactions among these node types.

propagate these signals without collapsing them during aggregation. This design prevents distinct relational structures from becoming indistinguishable after propagation and thereby improves the expressive power of the model.

## C.2 Connection with Meta Path

BG-HGNN addresses the challenges of parameter explosion and relation collapse by mapping diverse feature channels from various node types into a unified homogeneous space. However, whether BG-HGNN can automatically preserve and recognize the original heterogeneous information in this transformed space remains to be further examined. To validate this capability, we designed an experiment demonstrating that BG-HGNN can effectively discern meaningful patterns, particularly by identifying significant meta-paths without requiring dedicated weight spaces for each relation type. We examined the base model's activation/attention patterns against expert-defined meta-paths within the ACM dataset, which comprises Paper, Author, Subject, and Term nodes. The first hop interactions from expert-defined meta-paths are shown in Fig. 6(b), which are represented as binary matrices (1 for presence, 0 for absence). Upon model convergence, we analyzed the attention scores for each relationship, binarizing these scores with a threshold of 0.1. The results for the first hop interactions of our method are illustrated in Fig. 6(a). The resemblance between these matrices suggests that the baseline can inherently identify crucial meta-paths, highlighting our framework can indeed preserve meaningful information regarding the interactions among different relations.

Continuing this analysis, we delved into interactions spanning various hops, calculating the average attention scores across different relationships. The meta-paths emerging with the highest average scores included:

1. **Paper-Author-Paper (PAP)**: Connects two papers through a common author, reflecting potential similarities in content or research area.

2. **Paper-Subject-Paper (PSP)**: Links papers via their shared subject or research field, facilitating the identification of papers within related fields.

3. **Paper-Author-Paper-Author-Paper (PAPAP)**: Suggests similarities in research topics or fields by connecting papers through two authors.

These identified meta-paths correspond with those predefined by experts for the ACM dataset, reinforcing the notion that our framework is adept at capturing and refining the essence of interactions across different relations. Furthermore, the results indicate that even standard homogeneous GNNs, such as GAT, can inherently process and utilize heterogeneous data when facilitated by our method. This efficiency suggests

that the dependence on pre-established meta-paths and the use of diverse weight spaces for each relation may be superfluous, emphasizing the inherent capability of BG-HGNN to navigate and exploit the complex web of heterogeneous information effectively.

### C.3    Comparison with Relation-Specific Parameter Reduction Techniques

A natural approach to mitigating parameter growth in canonical HGNNs is to reduce the number of relation-specific parameters. Common techniques include (i) reducing the dimensionality of each relation-specific weight matrix, or (ii) representing relation-specific matrices as low-rank outer products of learnable vectors. While these methods can decrease the absolute number of parameters, they do *not* change the fundamental scaling behavior: the total parameter count still grows linearly with the number of relations $|\mathcal{R}|$, with only the constant factor reduced.

In contrast, our proposed BG-HGNN framework fundamentally removes the dependence on $|\mathcal{R}|$. By fusing heterogeneous node attributes, node-type encodings, and relation-type embeddings into a shared low-rank feature space, and applying shared-parameter message passing, BG-HGNN completely eliminates the need for distinct parameters per relation. This design ensures that the parameter complexity is independent of $|\mathcal{R}|$, enabling scalability to complex heterogeneous graphs with tens or hundreds of relation types, without sacrificing expressive power or predictive performance (Propositions 3.1 and 3.2).

While dimensionality reduction or low-rank approximations may lower memory footprint slightly, they cannot prevent the linear scaling of parameters with respect to the number of relations. BG-HGNN, by contrast, addresses the root cause of parameter explosion and provides both practical efficiency and theoretical guarantees on expressiveness.

## D    Experiment Details

This appendix provides additional details on the experiments discussed in the main text, including dataset information, the experimental setup, and specific implementation details. For each method introduced in the main text, we aim to include pseudocode and a thorough explanation to facilitate understanding and reproducibility.

### D.1    Datasets Description

Our experiments were designed to evaluate the model's performance on two fundamental tasks: node classification and link prediction. Below, we provide a detailed overview of the datasets used for these tasks.

### D.1.1    Node Classification Datasets

For the node classification task, we utilized eight distinct datasets. These datasets vary significantly in size, complexity, and the nature of the entities and relations they contain. The key statistics for each dataset are summarized in Table 3

### D.1.2    Link Prediction Datasets

For the link prediction task, we selected three datasets that offer a range of challenges, from predicting interactions between users and products to connections within social networks. The characteristics of these datasets are detailed in the table  4

### D.2    Experimental Environment

Our experiments were conducted on a Dell PowerEdge C4140, The key specifications of this server, pertinent to our research, include:
**CPU:** Dual Intel Xeon Gold 6230 processors, each offering 20 cores and 40 threads.
**GPU:** Four NVIDIA Tesla V100 SXM2 units, each equipped with 32GB of memory, tailored for NV Link.

**Memory:** An aggregate of 256GB RAM, distributed across eight 32GB RDIMM modules.
**Storage:** Dual 1.92TB SSDs with a 6Gbps SATA interface.
**Networking:** Features dual 1Gbps NICs and a Mellanox ConnectX-5 EX Dual Port 40/100GbE QSFP28 Adapter with GPUDirect support.
**Operating System:** Ubuntu 18.04LTS.

### D.3 Implementation

In our method, we begin by projecting the features of diverse node types into a unified feature space. This is achieved by amalgamating the attributes of all nodes to create a comprehensive common space, which encompasses all potential features. Attributes inherent to a node retain their original values, while absent features are assigned a default value, typically zero. This approach presents two significant benefits: firstly, it enables the sharing of weights across different node types, enhancing the efficiency of the learning process. Secondly, the incorporation of default values for missing features implicitly encodes information about the heterogeneity of node types, thus enriching the model's learning context. For additional details, see the pseudocode provided in Algorithm 1.

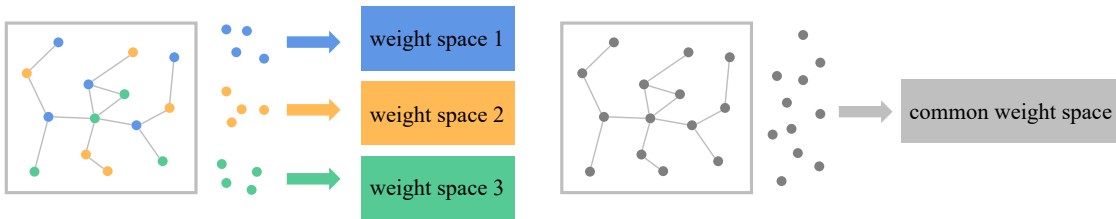

Figure 7: Transformation from a heterogeneous graph to a homogeneous graph, enabling nodes of diverse types to share the same weight space, thereby streamlining computation and promoting feature uniformity.

---

**Algorithm 1** Feature Space Projection

---

**Require:** A heterogeneous graph $\mathcal{G}$ with a node set $\mathcal{V}$ and associated feature sets $\mathcal{X}_v$ for each node $v$.
**Ensure:** Projected feature set $\mathcal{X}'_v$ in a unified feature space $\mathcal{X}_{\mathcal{T}_v}$ for each node $v$.
1: $\mathcal{X}_{\mathcal{T}_v} \leftarrow \bigcup_{\tau \in \mathcal{T}_v} \mathcal{X}_\tau$
2: **for** each node $v$ in $\mathcal{V}$ **do**
3:      Initialize $\mathcal{X}'_v$ to be a vector of size $|\mathcal{X}_{\mathcal{T}_v}|$ with all entries set to 0
4:      **for** each feature $i$ in $\mathcal{X}_{\mathcal{T}_v}$ **do**
5:          **if** feature $i$ exists in $\mathcal{X}_v$ **then**
6:              $\mathcal{X}'_v[i] \leftarrow \mathcal{X}_{v,i}$
7:          **end if**
8:      **end for**
9: **end for**

---

Subsequent to establishing a common feature space, our methodology seeks to incorporate the information concerning node types and relations from the heterogeneous graph into the node features. By doing so, even when the heterogeneous graph is translated into a homogeneous graph, the node features within the latter inherently encapsulate the heterogeneous information. To accomplish this integration, we begin by encoding the types of nodes and edges present in the heterogeneous graph. For this encoding, we adopt a strategy wherein each type is encoded as a random vector. Interestingly, our observations indicate that the optimal length of the random vector, which yields the best results, varies depending on the task at hand. Thus, we employ node type vectors to directly represent node type information. For relation information,

we aggregate the vectors of edges connected to each node and compute their mean to obtain a composite vector, effectively capturing the relational context of each node. The details are shown in Algorithm 2

---

**Algorithm 2** Encode Node types and Relations

---

**Require:** Graph $G$ with node set $V$, edge set $E$, node types $T_V$, and edge types $T_E$
**Ensure:** Node Type Features $\Omega_V$ and Relation Features $\Omega$
 1: Initialize empty dictionaries $\mathcal{O}_{\mathcal{T}_V}$ and $\mathcal{O}_{\mathcal{T}_E}$
 2: Initialize empty dictionary $\Omega$
 3: **for** each node type $\tau_v$ in $T_V$ **do**
 4:     $\mathcal{O}_{\mathcal{T}_V}[\tau_v] \leftarrow$ Vector in $\mathbb{R}^m$ with components drawn from $\mathcal{U}(a, b)$
 5: **end for**
 6: **for** each edge type $\tau_e$ in $T_E$ **do**
 7:     $\mathcal{O}_{\mathcal{T}_E}[\tau_e] \leftarrow$ Vector in $\mathbb{R}^n$ with components drawn from $\mathcal{U}(a, b)$
 8: **end for**
 9: **for** each node $v$ in $V$ **do**
10:     $\Omega_v \leftarrow \mathcal{O}_{\mathcal{T}_V}[\text{type of } v]$
11:     Initialize empty list ConnectedEdgeEncodings
12:     **for** each neighbor $u$ of $v$ **do**
13:         Append $\mathcal{O}_{\mathcal{T}_E}[\text{type of } (u, v)]$ to ConnectedEdgeEncodings
14:     **end for**
15:     $\Omega[v] \leftarrow$ Average of vectors in ConnectedEdgeEncodings
16: **end for**

---

Having executed the aforementioned steps, we are equipped with node information at three distinct levels: the intrinsic features which have been mapped into a common space $\mathcal{X}_{\mathcal{T}_V}$, the node type encoding $\mathcal{O}_{\mathcal{T}_V}$, and the relation encoding with all the neighbors $\Omega$. The pressing task at hand is to fuse these diverse features, thereby formulating the input for our homogeneous graph neural network. Departing from standard feature aggregation methods like vector concatenation, our methodology adopts the Kronecker product. Crucially, we append a '1' to each feature vector beforehand, a step that serves several purposes. It not only captures the interaction between different modalities, but also preserves the distinct features of each one. Here, we take a two-dimensional example as an illustration; the detailed principle can be seen in Fig 7.

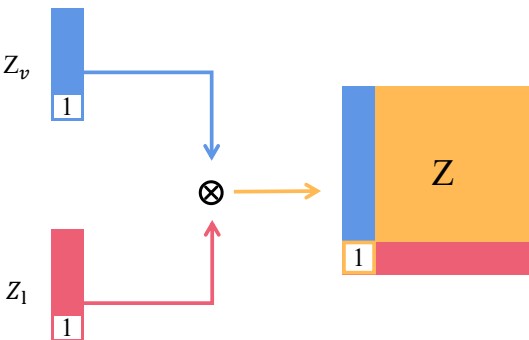

Figure 8: Illustration of Kronecker Product post appending '1' to feature vectors. The merged space in yellow represents the interactivity between the two modalities. Meanwhile, the distinct regions colored in red and blue encapsulate the inherent features of each individual modality.

Upon completing the initial processing, the three distinct features undergo a Kronecker product operation, subsequently passed through a linear layer, culminating in a singular one-dimensional feature vector. A crucial concern arising from the Kronecker product of three vectors is the potential exponential escalation

in the number of parameters. To strategically address this predicament, we draw inspiration from the Low-rank Multimodal Fusion (LMF) approach, which is shown in Fig 9. The essence of LMF lies in its ability to parallelly decompose tensors and weights. This decomposition leverages modality-specific low-rank factors to carry out the multimodal fusion. By eschewing the computation of high-dimensional tensors, this method not only curtails memory overhead but also adeptly mitigates the time complexity, transforming it from exponential to linear. The corresponding pseudocode is provided in Algorithm 3.

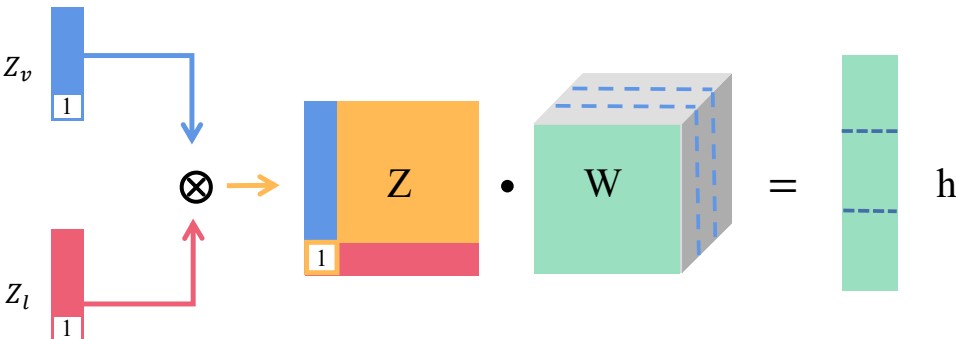

Figure 9: Visualization of weight decomposition leveraging the Low-rank Multimodal Fusion (LMF) approach. This showcases the breakdown of weights, sidestepping the necessity for expansive tensor operations and adeptly highlighting the strategy to circumvent the parameter explosion.

---

**Algorithm 3** Feature Fusion and Projection

---

**Require:** Node attribute vectors $\bar{\mathbf{x}}'_v$ of shape $[n, d_{\mathbf{x}}]$;
  1: Node type vectors $\mathbf{o}_{\tau_{v,i}}$ of shape $[n, d_{\mathbf{o}}]$;
  2: Neighbor relation vectors $\omega_v$ of shape $[n, d_\omega]$;
  3: Projection weights $\mathbf{W}$ for low-rank approximation with rank $r$
**Ensure:** Projected node features $\mathbf{h}_v$ of shape $[n, d_h]$
  4: Append a 1 to each vector in $\bar{\mathbf{x}}'_v$, $\mathbf{o}_{\tau_{v,i}}$, and $\omega_v$ to ensure inclusion of original and interaction features
  5: Initialize $\mathbf{h}_v$ as an empty vector for each node
  6: **for** each node $v$ in $V$ **do**
  7:     **for** $i \leftarrow 1$ to $r$ **do**
  8:         Extract rank-1 weight components: $\mathbf{w}_i^{(\mathbf{x})}$, $\mathbf{w}_i^{(\mathbf{o})}$, and $\mathbf{w}_i^{(\omega)}$
  9:         Compute partial projections for each feature type and rank:
 10:             $\mathbf{p}_{v,i}^{(\mathbf{x})} \leftarrow \mathbf{w}_i^{(\mathbf{x})} \cdot \bar{\mathbf{x}}'_v$
 11:             $\mathbf{p}_{v,i}^{(\mathbf{o})} \leftarrow \mathbf{w}_i^{(\mathbf{o})} \cdot \mathbf{o}_{\tau_{v,i}}$
 12:             $\mathbf{p}_{v,i}^{(\omega)} \leftarrow \mathbf{w}_i^{(\omega)} \cdot \omega_v$
 13:             $\mathbf{h}_{v,i} \leftarrow \mathbf{p}_{v,i}^{(\mathbf{x})} \odot \mathbf{p}_{v,i}^{(\mathbf{o})} \odot \mathbf{p}_{v,i}^{(\omega)}$
 14:             $\mathbf{h}_v \leftarrow \mathbf{h}_v + \mathbf{h}_{v,i}$
 15:     **end for**
 16: **end for**

---

### D.4 Hyperparameter Selection

We adopt a consistent and reproducible hyperparameter tuning protocol for BG-HGNN and all baseline methods. For each model, hyperparameters are selected based on validation performance using a comparable tuning budget.

**General setup.** For all methods, we tune the following common hyperparameters: learning rate, hidden dimension, dropout rate, and weight decay. We use the Adam optimiser with default setting and early stopping based on validation performance. Unless otherwise specified, we apply the same data splits and training procedures across all models to ensure fair comparison.

**BG-HGNN hyperparameters.** In addition to the common hyperparameters, BG-HGNN introduces the following model-specific parameters: (i) the rank $r$ in the low-rank interaction fusion module, and (ii) the dimension of the random type encodings. We search $r$ over a small range (e.g., $r \in \{1, 2, 3, 4, 5, 6, 7, 8, 9, 10\}$). Empirically, we find that a small constant rank ($r = 4$ or 5) provides a good trade-off between performance and efficiency. The encoding dimension is set proportional to the number of types (default $3 \times |T_V|$) with additional sensitivity analysis reported in Fig. 12.

**Baseline hyperparameters.** For baseline methods (e.g., RGCN, HGT, Simple-HGN, NARS, and Slot-GAT), we follow the recommended settings from their original papers or official implementations when available. We further tune key hyperparameters under the same validation protocol, including hidden dimension, dropout, and model-specific parameters such as the number of layers, attention heads, or slot numbers. In particular, we search hidden dimension over $\{64, 128, 256\}$, dropout over $\{0, 0.2, 0.5\}$, and model-specific parameters over small standard ranges (e.g., number of layers in $\{2, 3, 4\}$, attention heads in $\{2, 4, 8\}$, and slot numbers in $\{2, 4, 8, 16\}$), selecting the best configuration based on validation performance.

**Selection protocol.** All hyperparameters are selected based on the best validation performance, and the reported results correspond to the model configuration achieving the highest validation score. To account for randomness, we repeat each experiment five times with different random seeds and report the average performance.

### D.5  Additional Experiments

#### D.5.1  Additional Result on Link Prediction

We present additional results on the performance of our model compared to baselines on link prediction task. The result is presented at Table 5.

#### D.5.2  Preprocessing Time Analysis

To gain a deeper understanding of the computational costs associated with our method, we extend our analysis beyond the comparison of time required per epoch for different models in the main text. This extension involves considering the preprocessing time, encompassing essential tasks such as encoding node types and edge types, calculating node degrees, deriving node type features and relation features for each node, and mapping diverse node features into a common space. In this experiment, we adopt the execution time for a single epoch of the BG-HGNN model on the current dataset as the reference baseline, then we quantify the preprocessing time equivalently in terms of the number of epochs of BG-HGNN. For the sake of providing a more intuitive comparison, we chose to contrast the preprocessing time with the duration of a single epoch in the Heterogeneous Graph Transformer (HGT) model. The summarized results are presented in Fig 10.

The experiments demonstrate that the overall time required for BG-HGNN pretraining does not exceed the duration of a single epoch in the Heterogeneous Graph Transformer (HGT) model. This finding attests that the tensor-based fusion process introduces only negligible additional preprocessing time. Considering the substantial performance gains achieved in subsequent tasks, this incremental time overhead is deemed acceptable.

#### D.5.3  Efficacy of Encoding Node Types and Relations

To further validate the rationale behind embedding node features after encoding both node types and relations, we conducted an ablation study. In this experiment, on a previously utilized dataset, we compared

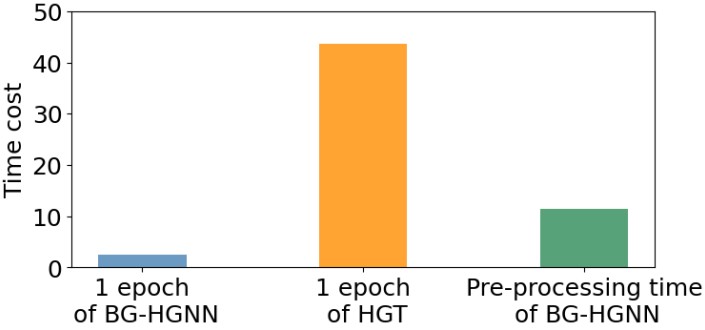

Figure 10: The plot showcases the time cost for essential preprocessing tasks in BG-HGNN including encoding node and edge types, computing node degrees, deriving node and relation features, and mapping node features into a common space. It is shown that the time cost of preprocessing is less than a single epoch duration of the Heterogeneous Graph Transformer (HGT) model.

the average accuracy of the BG-HGNN model under several configurations: using only the original features, using the cross-product of original features with node type encoding, using the cross-product of original features with relation encoding, and using the cross-product of original features with both node type and relation encoding. The results indicated that the BG-HGNN model, when integrating all three pieces of information, achieved the highest accuracy. This outcome further corroborates the efficiency of our method in preserving the original heterogeneous information within the transformed homogenous graph. The result is shown in Fig 11.

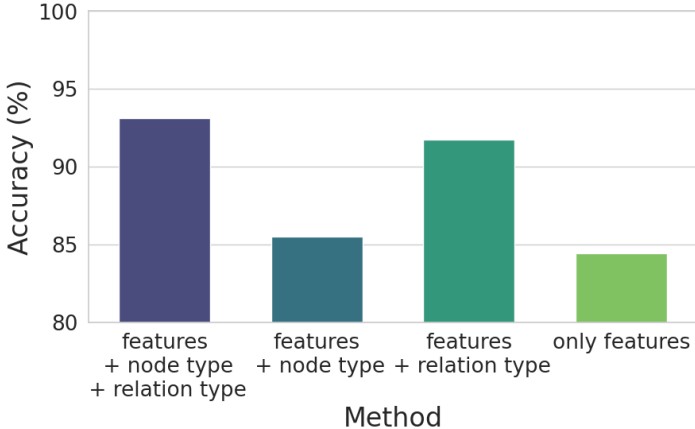

Figure 11: Accuracy comparison of BG-HGNN using various feature sets, demonstrating the enhanced performance when integrating node and relation type encodings with original features.

**Robustness of random dense encoding.** We further examine the robustness of the random dense encoding with respect to both the sampling process and the encoding dimension. To ensure that the reported results do not depend on a particular realization of the random encoding, we re-sample the node-type and relation-type encodings for each run using different random seeds. Table 6 reports the resulting performance variance across seeds. The standard deviations are consistently small, indicating that BG-HGNN is stable under different random encoding realizations and that its performance does not rely on a fortunate sampling instance. In addition, Fig. 12 studies the sensitivity of the model to the encoding dimension. The results show that when the encoding dimension is very small, performance can be less stable, as the random vectors may not provide sufficiently reliable separation between different types. However, once the encoding dimension

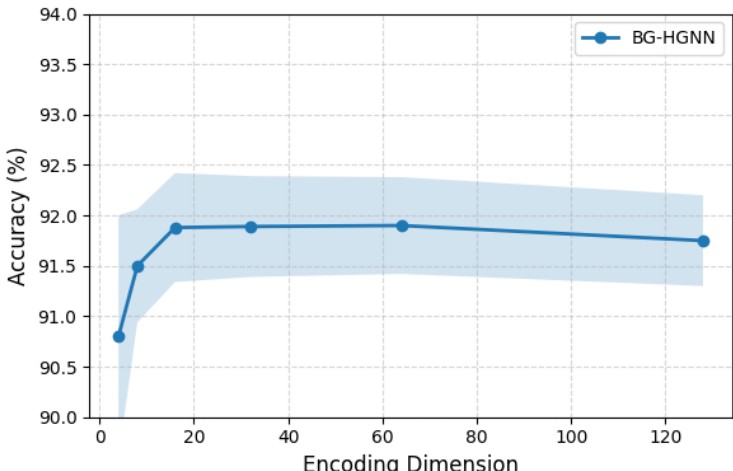

Figure 12: Performance of BG-HGNN under different encoding dimensions on the ACM dataset (8 relation types). The results show that the performance of BG-HGNN is largely insensitive to the encoding dimension once it exceeds a modest threshold. When the encoding dimension is very small, the random encoding may not provide sufficiently stable separation between types, resulting in slightly higher variance. However, as the dimension increases, the variance becomes negligible and the performance remains stable across different settings.

exceeds a modest threshold relative to the number of types/relations in the dataset, the performance stabilizes and becomes largely insensitive to further increases in dimension. These observations suggest that the gains of BG-HGNN arise from the proposed representation mechanism itself rather than incidental randomness, and are consistent with the theoretical intuition that sufficiently high-dimensional random encodings preserve type distinctions with high probability.

Table 2: Notation summary for BG-HGNN.

| Notation | Meaning |
|---|---|
| $G = (V, E, T_V, T_E, \phi, \psi)$ | Heterogeneous graph, where $V$ is the node set, $E$ is the edge set, $T_V$ is the set of node types, $T_E$ is the set of edge/relation types, $\phi : V \to T_V$ maps a node to its type, and $\psi : E \to T_E$ maps an edge to its type. |
| $v, u \in V$ | Nodes in the graph. |
| $(u, v) \in E$ | Edge between nodes $u$ and $v$. |
| $\tau_v \in T_V$ | Node type. In context, $\tau_v = \phi(v)$ denotes the type of node $v$. |
| $\tau_e \in T_E$ | Edge/relation type. In context, $\tau_e = \psi(u, v)$ denotes the type of edge $(u, v)$. |
| $x_v$ | Original feature vector of node $v$, defined on its type-specific feature space. |
| $\mathcal{F}_\tau$ | Set of feature channels associated with node type $\tau$. |
| $\mathcal{F}_{T_V} = \bigcup_{\tau \in T_V} \mathcal{F}_\tau$ | Global unified feature space formed by taking the union of feature channels over all node types. |
| $D_f = |\mathcal{F}_{T_V}|$ | Dimension of the unified feature space. |
| $\bar{x}_v \in \mathbb{R}^{D_f}$ | Padding-aligned feature vector of node $v$ in the unified feature space. |
| pad_val | Padding value used for feature channels not applicable to a given node type. |
| $h_v^{(\ell)}$ | Hidden representation of node $v$ at HGNN/GNN layer $\ell$. |
| $h_{v,r}^{(\ell)}$ | Intermediate representation of node $v$ associated with relation $r$ at layer $\ell$. |
| $\mathcal{N}_r(v)$ | Neighbors of node $v$ connected through relation type $r$. |
| $\mathcal{N}(v)$ | Full neighborhood of node $v$. |
| $z_{\tau_v} \in \mathbb{R}^{D_{\tau_v}}$ | Random dense encoding of node type $\tau_v$. |
| $D_{\tau_v}$ | Dimension of node-type encoding. The paper uses $D_{\tau_v} = 10|T_V|$. |
| $\mathcal{Z}_{T_V} = \{z_\tau \mid \tau \in T_V\}$ | Collection of all node-type encoding vectors. |
| $o_{\tau_e} \in \mathbb{R}^{D_{\tau_e}}$ | Random dense encoding of edge/relation type $\tau_e$. |
| $D_{\tau_e}$ | Dimension of edge/relation encoding. |
| $\omega_v$ | Aggregated edge-type encoding for node $v$, computed by averaging encodings of incident edge types in its neighborhood. |
| $\Omega = \{\omega_v \mid v \in V\}$ | Collection of node-wise aggregated edge-type encodings. |
| $H_v = \bar{x}_v \otimes z_{\tau_v} \otimes \omega_v$ | Conceptual full Kronecker-product interaction tensor for node $v$, capturing all multiplicative cross-modal interactions. |
| $\otimes$ | Kronecker product. |
| $\odot$ | Element-wise product. |
| $r$ | Rank hyperparameter in the low-rank interaction fusion module. |
| $D_h$ | Output dimension of the fused representation. |
| $w_i^{(x)} \in \mathbb{R}^{D_h \times D_f}$ | Learnable projection parameter for aligned node attributes in rank component $i$. |
| $w_i^{(z)} \in \mathbb{R}^{D_h \times D_{\tau_v}}$ | Learnable projection parameter for node-type encoding in rank component $i$. |
| $w_i^{(\omega)} \in \mathbb{R}^{D_h \times D_{\tau_e}}$ | Learnable projection parameter for aggregated edge-type encoding in rank component $i$. |
| $h_v \in \mathbb{R}^{D_h}$ | Final fused node representation produced by low-rank interaction fusion before shared-parameter message passing. |
| $L$ | Number of GNN/HGNN layers. |
| $D$ | Hidden dimension used in the theoretical parameter-complexity analysis. |
| $W_r^{(\ell)} \in \mathbb{R}^{D \times D}$ | Relation-specific weight matrix for relation $r$ at layer $\ell$ in canonical HGNNs. |
| $\mathcal{M}_{\text{HGNN}}$ | Model class of canonical HGNNs with relation-wise mean/sum aggregation. |
| $\mathcal{M}_{\text{BG-HGNN}}$ | Model class of BG-HGNNs. |
| $\succeq, \succ$ | Relative expressiveness relations between model classes: "at least as expressive as" and "strictly more expressive than." |
| $G_1, G_2, G_3, G_4$ | Heterogeneous subgraphs used in the expressiveness definition and proofs. |
| $m_r(v)$ | Mean of neighbor features of node $v$ under relation $r$ in the relation-collapse proof. |
| $\mu_i$ | Mean of the Gaussian neighbor-feature distribution associated with relation type $r_i$ in Lemma A.1. |
| $\sigma(\cdot)$ | Pointwise nonlinearity. |

| Dataset | Entities | Relations | Edges | Labeled | Classes |
|---|---|---|---|---|---|
| AM | 1666764 | 133 | 5988321 | 1000 | 11 |
| AIFB | 8285 | 45 | 29043 | 176 | 4 |
| MUTAG | 23644 | 23 | 74227 | 340 | 2 |
| BGS | 333845 | 103 | 916199 | 146 | 2 |
| DBLP | 26128 | 6 | 239566 | 4057 | 4 |
| IMDB | 21420 | 6 | 86642 | 4573 | 5 |
| ACM | 10942 | 8 | 547872 | 3025 | 3 |
| Freebase | 180098 | 36 | 1057688 | 7954 | 7 |

Table 3: Summary of datasets used in the node classification task. The table provides information on the number of entities, relations, edges, labeled entities, and classes for each dataset.

| Dataset | Entities | Node Types | Edges | Relations | Target |
|---|---|---|---|---|---|
| Amazon | 10099 | 1 | 148659 | 2 | product-product |
| LastFM | 20612 | 3 | 141521 | 3 | user-artist |
| Youtube | 1999 | 1 | 1179537 | 4 | user-user |

Table 4: Summary of datasets used in the link prediction task. This table outlines the number of entities, node types, edges, relations, and the prediction targets for each dataset.

| 2-hop node pairs test | ROC-AUC↑ | MRR↑ | ROC-AUC↑ | MRR↑ | ROC-AUC↑ | MRR↑ |
|---|---|---|---|---|---|---|
| Model | LastFM | | Amazon | | Youtube | |
| HGT | 55.68 | 75.56 | 91.31 | 95.80 | - | - |
| RGCN | 57.22 | 77.60 | 84.18 | 93.10 | 69.91 | 88.58 |
| simple-HGN | 60.18 | 73.61 | 76.89 | 90.74 | 67.35 | 78.21 |
| NARS | 59.25 | 78.44 | 85.62 | 93.24 | 70.24 | 85.62 |
| slotGAT | 60.20 | 76.77 | 89.23 | 95.78 | 72.42 | 90.16 |
| BG-HGNN | 57.04 | 80.37 | 89.40 | 97.71 | 86.72 | 95.07 |
| random-hop node pairs test | ROC-AUC↑ | MRR↑ | ROC-AUC↑ | MRR↑ | ROC-AUC↑ | MRR↑ |
| Model | LastFM | | Amazon | | Youtube | |
| HGT | 80.45 | 95.81 | 96.61 | 98.39 | - | - |
| RGCN | 82.05 | 96.54 | 87.63 | 93.10 | 77.76 | 88.28 |
| simple-HGN | 81.38 | 95.13 | 80.41 | 91.91 | 67.44 | 78.67 |
| NARS | 82.03 | 96.22 | 96.44 | 95.57 | 79.65 | 92.42 |
| slotGAT | 85.68 | 97.22 | 96.82 | 97.63 | 80.23 | 91.14 |
| BG-HGNN | 87.45 | 97.80 | 97.71 | 99.18 | 86.86 | 95.69 |

Table 5: Comparative analysis across different datasets in link prediction task. The performance is evaluated using ROC-AUC and MRR metrics. The best performance is marked in green and the second best performance is marked in light green. The top part of the table shows the results for link prediction tested on 2-hop node pairs, while the bottom part presents the results on random hop node pairs. The reported results are the average of five independent trials. Entries with "-" indicate the experiment instance is over of memory in our testbed.

Table 6: Performance variance of BG-HGNN across random seeds for random dense encoding. Each run re-samples node-type and relation-type encodings with a different seed.

| Dataset | Mean Accuracy | Std |
|---|---|---|
| ACM | 91.88 | 0.48 |
| AIFB | 97.22 | 0.56 |
| MUTAG | 83.60 | 0.94 |
| BGS | 94.49 | 0.77 |

