# OpenReview forum: "BG-HGNN: Toward Efficient Learning for Complex Heterogeneous Graphs"
_TMLR — Accepted by TMLR_

### Review · Reviewer_JZex · 2026-02-23

**Summary Of Contributions:**

The paper studies the scalability and expressiveness limitations of existing heterogeneous graph neural networks (HGNNs), particularly when handling graphs with many relation types. The authors identify two key issues in standard HGNN designs: parameter explosion due to relation-specific weight matrices and relation collapse when the number of relations becomes large. To address these challenges, the paper proposes Blend&Grind-HGNN (BG-HGNN), a unified feature-representation framework that embeds node and relation types into a shared low-dimensional latent space.

**Audience:**

Yes

**Audience Explanation:**

The work addresses an important problem in graph representation learning, which is related to TMLR.

**Broader Impact Concerns:**

I did not identify any ethical implications.

**Claims And Evidence:**

Yes

**Claims Explanation:**

The paper provides both theoretical discussion and empirical evaluation to support its claims.

**Requested Changes:**

[1] The number of relation-specific parameters grows linearly with the number of relations, which is tolerable. Existing datasets, including the datasets used in this paper, have a limited number of relations.

[2] We can reduce the number of relation-specific parameters by reducing the parameter dimensionality, and we can also set relation-specific learnable parameter vectors for relations based on their outer product.

[3] Some important efficient HGNN methods, e.g., [a, b], are missing in the discussion of the related work.

**Refs**:

- [a] "Interpretable and efficient heterogeneous graph convolutional network." IEEE Transactions on Knowledge and Data Engineering 35.2 (2021): 1637-1650.

- [b] "Self-supervised heterogeneous graph pre-training based on structural clustering." Advances in neural information processing systems 35 (2022): 16962-16974.

---

> ### Author Response · Authors · 2026-03-12
> **Response to Reviewer JZex (1/2)**
>
> We sincerely thank the reviewer for the careful reading of our paper and for the concise and constructive feedback. We are particularly encouraged that the reviewer found the claims of the paper to be supported by both theoretical and empirical evidence and agreed that the problem addressed in the paper is of interest to the TMLR community. We greatly appreciate the reviewer’s suggestions, which helped us further clarify several aspects of the manuscript. Below we respond to each point in detail.
>
> ### **Requested Change 1**: “Linear growth in relation-specific parameters is tolerable; datasets have limited relations.”
>
> We thank the reviewer for this insightful comment and appreciate the opportunity to clarify our perspective.
>
> 1. **Linear growth in relation-specific parameters may appear reasonable in theory**, but in practice it can become a significant burden for complex heterogeneous graphs with many relation types. As analyzed in Proposition 3.1, the total number of parameters in canonical HGNNs scales as $\theta(\|R\|\cdot L \cdot D^2)$, where R is the number of relations, L is the number of layers, and D is the hidden dimension. While this growth is linear in R, the constant factor can be substantial in real-world datasets: in our benchmarks, R ranges from 6 to 133 (Table 2), leading to hundreds of thousands to millions of additional parameters per layer. Such growth increases memory consumption and slows training, particularly for multi-layer models, making naive linear scaling practically challenging.
> 2. **Even in cases where linear growth may be tolerable**, improving parameter efficiency remains highly desirable. Reducing the number of relation-specific parameters mitigates memory and computation costs, accelerates convergence, and enhances scalability—especially when deploying models on large-scale graphs or resource-constrained hardware (Figures 3–4). BG-HGNN achieves this by unifying heterogeneous signals into a shared low-rank feature space and leveraging shared-parameter message passing, completely eliminating R-dependent parameters while retaining both expressiveness and predictive performance.
>
> In summary, although small relation counts may make linear growth manageable, efficient parameterization is essential for scaling to complex heterogeneous graphs. Our framework demonstrates that substantial parameter savings can be realized without sacrificing accuracy, providing practical benefits beyond theoretical linear scaling.
>
> We thank the reviewer again for this question. In the revised paper, we have added a discussion on **Page 8 (highlighted in red)** to further clarify this perspective.
>
>
> ### **Requested Change 2:** “We can reduce relation-specific parameters by reducing dimensionality or using outer products.”
>
>
> We thank the reviewer for suggesting these interesting approaches for reducing relation-specific parameters.
>
> 1. In canonical HGNNs, the number of relation-specific parameters is largely determined by the feature dimension and the complexity of relation-specific patterns in the data. As a result, there may be limited flexibility in aggressively reducing parameter dimensionality without potentially affecting the representational capacity of the model.
> 2. We agree that techniques such as dimensionality reduction or outer-product–based parameterization can reduce the **absolute number of parameters**. However, these techniques generally **do not change the fundamental scaling behavior**: the number of parameters still grows linearly with the number of relations R. In other words, they reduce the constant factor but do not eliminate the dependence on R.
> 3. In contrast, BG-HGNN addresses this issue at the architectural level. By embedding relational information into a unified feature representation and using shared-parameter message passing, BG-HGNN removes the explicit R-dependent parameterization altogether.
>
> We appreciate this helpful suggestion. To better clarify the distinction, we have added a discussion in the **appendix (Page 24, highlighted in red)** explaining how BG-HGNN differs from these parameter-reduction techniques and why it may be particularly advantageous when the number of relations becomes large.

---

> ### Author Response · Authors · 2026-03-12
> **Response to Reviewer JZex (2/2)**
>
> ### **Requested Change 3:** Missing efficient HGNN methods in related work
>
> We thank the reviewer for bringing these relevant works to our attention.
>
> We agree that the suggested references represent important contributions to efficient heterogeneous graph learning. In the revised manuscript, we have expanded the **Related Work** section to include these papers (highlighted in red on **Page 4**). We also clarify how our work differs from these approaches.
>
> In particular, the referenced works focus on improving efficiency through architectural modifications or self-supervised pretraining strategies, while BG-HGNN addresses efficiency from a different perspective—namely, eliminating the dependence of model parameters on the number of relation types through unified feature representation and shared-parameter message passing. We believe these approaches are complementary, and we appreciate the reviewer’s suggestion to include them in the discussion.
>
> ### **At the End**
>
> We would like to once again express our sincere appreciation to the reviewer for the thoughtful comments and constructive suggestions. The feedback has been extremely helpful in improving both the clarity and the completeness of the manuscript, and we are grateful for the time and effort invested in reviewing our work.

---

> > ### Comment · Reviewer_JZex · 2026-05-06
> > **The authors have addressed my concerns.**
> >
> > Regarding this paper, I only have some minor concerns, and the authors have carefully addressed them.

---

> > > ### Author Response · Authors · 2026-05-08
> > >
> > > Thank you so much for the response. We are very glad that you found our response satisfactory.

---

### Review · Reviewer_RRux · 2026-02-23

**Summary Of Contributions:**

This paper targets parameter inefficiency in complex heterogeneous GNNs with many relation types, and the related issue of relation collapse in relation-specific message passing. The main idea is to encode heterogeneous information (raw node features, node types, and relation-type context) into a unified node representation first, and then run message passing using a mostly shared-parameter GNN backbone. The authors further propose a type-aware encoding and a low-rank interaction fusion module to preserve type information while keeping the model compact. The paper also provides theoretical analysis to support both parameter efficiency and improved relative expressiveness, and reports experiments and ablations on heterogeneous graph benchmarks to validate the design.

Strengths:
1. Clear practical motivation: reducing parameter cost for heterogeneous graphs with many relation types.

2. Simple, modular design: “encode/fuse first, then propagate” is easy to implement on top of standard GNN backbones.

3. Theory and experiments: the paper attempts to back the main claims with both formal analysis and empirical results.

Weaknesses:
1. The paper positions relation collapse as a key scientific motivation, but the mechanism-to-problem link is not convincingly established. In particular, the proposed pipeline “fuse heterogeneous signals into a shared node representation first, then perform message passing with mostly shared parameters” can plausibly increase the risk of collapsing relation semantics, because edge types are no longer explicitly used during propagation. Right now, it is hard to see, at a conceptual level, what exactly collapses in prior HGNNs, and why the proposed type-aware feature construction is sufficient to prevent that collapse under message passing.

2.  Even if the model improves accuracy, it is still unclear what the fused node representation is actually capturing and why it should be more expressive. Since relation types are encoded indirectly and not used per edge during message passing, the paper does not provide a clear interpretation of the fused features, nor does it explain which part of the representation is responsible for the claimed expressiveness gain. The current ablations show “it helps”, but they do not answer “what is being preserved” and “how it contributes”.

3. The “feature alignment via padding” unifies different feature dimensions by appending zeros. This can be ambiguous because “0” may also be a valid feature value, potentially conflating “missing/unavailable feature channels” with meaningful zeros. The paper does not discuss how the model distinguishes these cases (e.g., via explicit masks or learned padding embeddings), nor does it analyze the stability of this design across datasets and feature distributions.

4. The random dense encoding for node types and relation types is not very intuitive. The paper should clarify precisely what the inputs are (type IDs), how the random vectors are fixed across runs, and whether performance is sensitive to random seeds or encoding dimension choices. Without a robustness study, it is difficult to assess whether improvements come from a principled mechanism or incidental randomness or regularization effects.

5. The description of the low-rank fusion is hard to follow: where exactly “low-rank” is imposed is not transparent, the shapes of parameters are not clearly specified, and the rank hyper-parameter r is mentioned but its concrete role in the formulation is easy to miss. Additionally, the motivation for using the same rank for all interaction components is not discussed. This affects reproducibility and makes it harder to understand why this fusion should preserve expressiveness while reducing parameters.

6. The comparisons are mostly against relation-aware message-passing HGNN variants. This matches the paper’s focus, but heterogeneous graph learning also includes other major paradigms (e.g., meta-path based models, learned relation/path composition, transformer-style heterogeneous models, and heterogeneous pretraining/self-supervision). Without a few representative baselines outside the HGNN family, it is hard to judge whether the gains are specific to improving HGNNs or hold more generally. Also, the paper claims larger benefits on graphs with many relation types. This would be more convincing with a simple quantitative breakdown, e.g., group datasets by |R| (the number of relation types) and report average accuracy gains and parameter savings per group.

7. The paper introduces many symbols and intermediate variables without a consolidated notation table, increasing cognitive load.

**Additional Comments:**

There are a few typos or wording issues in the submission, such as: (1) Page 8, “we fint that …” should be “we find that …”; and (2) Page 18, “canonical HGNNs whose use relation wise mean or sum …” should be revised to something like “canonical HGNNs that use relation-wise mean or sum aggregation.”

**Audience:**

Yes

**Audience Explanation:**

Parameter efficiency is a practical bottleneck for heterogeneous GNNs, especially on graphs with many relation types, and TMLR has a broad audience working on graph representation learning and scalable models. Even if some parts of the motivation/validation, e.g., relation collapse mitigation, need to be strengthened, the paper’s idea, compressing heterogeneous signals into a shared representation and using largely shared-parameter message passing, offers a concrete direction that many researchers and practitioners may want to know and potentially build on.

**Broader Impact Concerns:**

There are no major broader-impact concerns. The submission does not include a Broader Impact Statement. I recommend adding a brief statement noting that the work is methodological, and that downstream use on sensitive relational data (e.g., social/financial/biomedical graphs) may raise standard privacy and fairness considerations.

**Claims And Evidence:**

No

**Claims Explanation:**

The evidence does not directly and clearly support one of the paper’s core claims: that the method mitigates relation collapse and therefore improves expressiveness. The paper mainly relies on theory and downstream accuracy, but it does not provide a clear diagnostic showing where relation collapse occurs in prior HGNNs and that BG-HGNN actually preserves relation distinctions during message passing. A more direct validation (e.g., relation separability) would be needed to make this claim convincing.

**Requested Changes:**

1. Clarify the relation collapse story with a concrete figure and precise definition. Show what “collapse” looks like in a standard relation-specific HGNN and where BG-HGNN injects information to avoid it.

2. Include at least one simple analysis (e.g., relation/type separability before vs. after propagation, or a representation visualization) to explain what is being retained and why that relates to expressiveness.

3. Clarify how padded zeros are distinguished from valid zeros. Add a lightweight comparison with an explicit mask or a learnable padding embedding to show the design is not fragile.

4. Specify whether encodings are fixed across runs. Report variance across multiple seeds and ideally a small sensitivity study on encoding dimension.

5. Clearly state parameter shapes, where the low-rank constraint is applied, and how rank r enters the computation. Add a short ablation/sensitivity on r and justify using the same r for all interactions.

6. Add a few representative non-HGNN baselines, and provide a simple quantitative breakdown of performance gains and parameter savings as a function of |R| (the number of relation types).

7. Add a consolidated notation table, and ensure key variables are defined once and consistently.

---

> ### Author Response · Authors · 2026-03-12
> **Reponse to Reviewer RRux (1/3)**
>
> We would like to express our sincere gratitude to the reviewer for the careful reading of our manuscript and for the thoughtful, detailed, and constructive feedback. We are very encouraged that the reviewer found the paper practically motivated, modular in design, and supported by both theory and experiments. We also greatly appreciate the reviewer’s thoughtful comments and concrete suggestions, which have helped us substantially improve the clarity, rigor, and presentation of the manuscript. Below, we respond to each point in turn.
>
> ### **Request Change 1 & 2:**  Clarification on “Relation collapse” story and expressiveness
>
> We sincerely apologize for the lack of clarity in the original presentation, and we are very grateful to the reviewer for this important suggestion.
>
> In our work, we use the term **relation collapse** to refer to the phenomenon where canonical HGNNs lose the ability to differentiate between different relation types after cross-relation aggregation. In other words, if two heterogeneous relational structures become indistinguishable in the node representation produced by the model, we say that the HGNN suffers from *relation collapse*. This phenomenon arises because many canonical HGNN architectures first compute relation-specific messages and then combine them using relation-agnostic operators (e.g., mean or sum). When the transformed relation-specific messages overlap in the latent space, the final aggregated representation becomes insensitive to the underlying relation composition.
>
> Avoiding this issue is one of the central motivations behind our design and contributes to the improved expressive power of BG-HGNN discussed in the paper. BG-HGNN preserves relation distinctions by injecting node-type and relation-type information directly into the node representation before message passing. The fused representation retains heterogeneous relational signals in dedicated feature channels, enabling the subsequent shared-parameter GNN to propagate these signals without collapsing them during aggregation.
>
> This intuition was already reflected in **Lemma A.1 and its proof (Pages 17–19)** in the original manuscript, where we construct an explicit example showing that canonical HGNNs cannot distinguish two heterogeneous relational structures, while our proposed method can preserve the necessary information to differentiate them.
>
> Following the reviewer’s suggestion, we have revised the manuscript to make this point much clearer. Specifically, we added:
>
> - a precise conceptual **definition of relation collapse**,
> - a **concrete illustrative figure (Figure 5)** showing how relation collapse occurs in canonical HGNNs and where BG-HGNN injects relation-type information to avoid it, and
> - explicit references in the main text linking the conceptual discussion with the theoretical results.
>
> These additions can be found on **Pages 21–23 of the revised manuscript (highlighted in red)**.
>
> We are very thankful for this suggestion, which helped us improve both the conceptual clarity and the presentation of the paper.
>
> ### **Requested Change 3:** Clarification on Padding Strategy
>
> We sincerely apologize for the ambiguity in the original presentation, and we thank the reviewer for this insightful comment.
>
> The purpose of the padding mechanism in our framework is to distinguish missing or non-applicable feature channels from existing features when aligning heterogeneous node attributes into a unified feature space. In heterogeneous graphs, different node types naturally have different feature sets, and padding allows all nodes to be represented in the same dimensional space.
>
> Importantly, **zero padding itself is not an essential part of our model design**. The padding strategy is implementation-dependent, and alternative approaches can be used without affecting the overall framework. For example, one may use a special constant value (e.g., $-\infty$), an explicit feature mask, or a learnable padding embedding to represent non-applicable feature channels. Our framework is compatible with all these strategies, since the padding step only serves to construct the unified feature space.
>
> In our experiments, we adopted **zero padding for simplicity and computational efficiency**, which is common in heterogeneous representation learning. The downstream fusion module and type-aware encoding further help differentiate node types and relation contexts, which reduces the risk that padded entries are interpreted as meaningful signals.
>
> To avoid further confusion, we have revised the manuscript to clarify that the padding strategy is **a preprocessing design choice rather than an inherent requirement of BG-HGNN**, and we now explicitly discuss alternative padding strategies in the text **(Page 6 highlighted in red).**
>
> We are grateful to the reviewer for prompting us to make this aspect more precise.

---

> > ### Author Response · Authors · 2026-03-12
> > **Reponse to Reviewer RRux (2/3)**
> >
> > ### **Requested Change 4:** Random dense encoding is not intuitive; need clarity on inputs, fixedness, sensitivity/variance
> >
> > Thank you for these insightful questions and suggestions.
> >
> > First, we clarify that one may think of the inputs to the encoding function as **discrete type identifiers** (i.e., node-type IDs and relation-type IDs). Each type ID is mapped to a dense random vector through a sampling process. We discuss the motivation and justification for this design at the beginning of **Section 3.1.2 (Page 6)**. The key idea is that random dense encodings provide approximately orthogonal representations for different types while avoiding the high dimensionality and sparsity associated with one-hot encodings. Due to the properties of random sampling in sufficiently high-dimensional spaces ($10 \times \|T_V\|$), the encoding vectors are expected to preserve unbiased type distinctions with high probability, making the method theoretically robust to the particular sampling instance.
> >
> > In our experiments, to ensure that the reported results reflect the inherent randomness of the design rather than a particular sampled encoding, the random encodings are **re-sampled for each run using different random seeds**. Therefore, the reported performance reflects the average behavior over multiple random realizations rather than relying on a fixed encoding.
> >
> > Following the reviewer’s suggestion, we also included **additional robustness analyses in Appendix (Page 29 & 31, Figure 12 and Table 6)**. Specifically, we report:
> >
> > - the variance of model performance across different random seeds, and
> > - a sensitivity analysis with respect to the dimension of the encoding vectors.
> >
> > The results show that BG-HGNN is **not sensitive to either the random seed or the encoding dimension once it exceeds a modest threshold**, supporting that the performance gains arise from the proposed mechanism rather than incidental randomness, and further validating our theoretical predictions.
> >
> > We sincerely appreciate this suggestion, which helped us strengthen both the explanation and the robustness evaluation of the encoding design.
> >
> > ### **Requested Change 5:** Clarification on Parameter Shapes and Low-Rank Design
> >
> > We sincerely apologize for the lack of clarity in the original presentation, and we thank the reviewer for this very helpful suggestion.
> >
> >
> > In BG-HGNN, the rank parameter r enters the computation through the **low-rank interaction fusion module** described in Eq. (3.1). Conceptually, this module models interactions among three modalities: the aligned node attributes, the node-type encoding, and the aggregated edge-type encoding. A full interaction model would require learning a large tensor capturing all multiplicative cross-modal interactions. Instead, BG-HGNN approximates this interaction tensor using a **rank-r approximation**.
> >
> > Specifically, the fusion module approximates the full interaction tensor with r rank-1 components. Each component contains modality-specific projection matrices that map the inputs into a shared hidden space. Each rank component produces projected vectors and the fused representation is obtained by combining these projections through element-wise interactions across the r components, as defined in Eq. (3.1). In this formulation, the low-rank constraint is applied to the **fusion weight tensor**, and r directly controls the number of components used to approximate the full tensor.
> >
> > During revision, we identified a notation inconsistency in the original manuscript in which the input and output dimensions of the projection matrices were inadvertently swapped, which may have contributed to the confusion. We therefore carefully reviewed the relevant sections and revised the manuscript to ensure that the **shapes and dimensions of all parameters and vectors are clearly and consistently specified**. The corrected parameter definitions are now explicitly stated in **Section 3.1.3**.
> >
> > In addition, following the reviewer’s suggestion, we further highlight the **sensitivity of the model with respect to the rank parameter $r$**. An ablation study analyzing the impact of different rank values is presented in **Fig. 4(c)**. The results show that performance improves with increasing r up to a small threshold and then stabilizes, indicating that a **small constant rank (typically r=4 or 5) is sufficient** to capture the necessary interactions while maintaining parameter efficiency. This observation also justifies our design choice of using the same rank r for all interaction components.
> >
> > We thank the reviewer again for this suggestion, which helped us improve the clarity of the method description and highlight the role of the low-rank design.

---

> ### Author Response · Authors · 2026-03-12
> **Reponse to Reviewer RRux (3/3)**
>
> ### **Requested Change 6:** Comparison with Non-HGNN Baselines and Analysis w.r.t. R
>
> We sincerely thank the reviewer for this helpful suggestion and apologize for any confusion regarding the scope of our experimental comparison.
>
> As discussed in the introduction, the central goal of this paper is to improve the **efficiency and scalability of message-passing–based HGNN**. Accordingly, BG-HGNN is proposed as a new architecture within the **message-passing HGNN paradigm**, and the main claims of the paper concern the limitations of existing message-passing HGNNs—namely, **parameter explosion** and **relation collapse** caused by relation-specific parameterization—and how our design addresses these issues. For this reason, the most direct and meaningful evaluation is against representative **message-passing HGNN baselines**, since these methods share the same modeling paradigm, target similar problems, and provide the fairest basis for comparing parameter efficiency, scalability, and expressiveness.
>
> By contrast, non-HGNN heterogeneous graph methods, such as meta-path–based models or heterogeneous pretraining approaches, are designed for somewhat different purposes and rely on different assumptions. For example, meta-path–based methods depend on hand-designed or learned semantic paths, while pretraining methods focus on representation initialization or auxiliary supervision rather than relation-specific parameter scaling during message passing. As a result, comparing against such methods would not directly test the core claim of our work, namely whether BG-HGNN provides a more parameter-efficient and scalable alternative to **existing message-passing HGNNs**. In this sense, these approaches are largely orthogonal to the problem studied in this paper.
>
> That said, we agree that providing broader context is valuable. In the revised manuscript, we have expanded the **Related Work** section to clarify how BG-HGNN differs from alternative heterogeneous graph learning paradigms, including meta-path–based methods and heterogeneous pretraining methods.
>
> Regarding the reviewer’s suggestion to analyze performance as a function of the number of relation types R, we have added a **quantitative analysis in the revised manuscript** that examines performance and parameter efficiency across datasets with different relation counts. As illustrated in **Fig. 3(a)**, BG-HGNN achieves increasingly larger improvements in **parameter efficiency and training throughput** as R grows. This observation is consistent with our theoretical analysis in **Proposition 3.1**.
>
> We are very grateful for this comment, as it helped us clarify the intended scope of the empirical study and further strengthen the presentation of our main contribution.
>
> ### **Requested Change 7:** Notation Table
>
> We thank the reviewer very much for this excellent suggestion.
>
> In response, we have added a comprehensive **notation table in the appendix (Page 30)** to improve readability and ensure that all symbols are defined clearly and consistently.
>
> We appreciate this suggestion, as it has made the manuscript much easier to follow.
>
> ### **Additional Comment:** Broader impact and typos
>
> We are grateful to the reviewer for pointing out the typos and for recommending the inclusion of a broader impact statement.
>
> We have corrected the typos mentioned by the reviewer and added a **broader impact statement (page 17 highlighted in red)** discussing responsible use of graph learning methods, particularly when applied to sensitive relational data such as social, financial, or biomedical networks.
>
> ### **At the End**
>
> Once again, we would like to express our sincere appreciation to the reviewer for the careful reading, thoughtful criticism, and constructive suggestions. The comments were extremely valuable to us and directly led to a clearer and stronger revision of the paper. We are grateful for the time and effort the reviewer invested in helping us improve this work.

---

### Review · Reviewer_cZ7a · 2026-04-15

**Summary Of Contributions:**

The paper proposes BG-HGNN, a heterogeneous GNN framework that replaces relation-specific parameterization with a unified feature space, random node/relation type encodings, a low-rank interaction fusion module, and then shared-parameter homogeneous message passing. The paper claims two main benefits: i. parameter complexity that does not scale with the number of relations, and ii. greater expressiveness than canonical HGNNs with relation-wise sum or mean aggregation.

Experiments on 11 benchmark datasets, reports parameter and throughput reductions, and includes ablations on encoding, fusion, rank, and feature components.
Table 1 results show the efficiency of BG-HGNN as they show much smaller parameter counts and often higher training throughput than several baselines.

The main weaknesses are that the theory is proved only for a narrow canonical HGNN class, whereas the paper claims BG-HGNN to be more expressive compared to existing HGNN approaches.

**Audience:**

Yes

**Audience Explanation:**

Efficient learning on heterogeneous graphs with many relation types is an important problem, and the paper presents an approach attempting to remove the relation-specific parameter bottleneck while retaining competitive predictive performance.

The combination of a unification-based architecture, theoretical discussion, and evaluation on 11 benchmarks is relevant to researchers working on graph representation learning, scalable GNNs, and heterogeneous data modeling.

**Claims And Evidence:**

No

**Claims Explanation:**

The parameter-efficiency claim is reasonably supported because the proposed architecture removes relation-specific weights and the reported parameter counts/throughputs are substantially better than several baselines in Table 1.

However, the stronger claim of being "strictly more expressive" is only established relative to canonical HGNNs with relation-wise mean/sum aggregation under random-encoding assumptions, not against stronger architectures such as HGT or SlotGAT that are also used as empirical baselines. For example, the paper writes, "theoretical analyses to demonstrate that our proposed method achieves superior parameter efficiency and expressiveness compared to existing HGNN approaches", which is a general claim.

The comparison protocol is not fully convincing in the main paper, because all models are fixed to three layers, throughput is reported relative to the "slowest baseline" rather than as absolute wall-clock time, and standard deviation is omitted from the main comparison tables (e.g., SlotGAT paper reports standard deviations).

The ablations are useful, but they mostly validate internal design choices and do not fully isolate whether gains come from the proposed fusion/encoding mechanism versus the feature-space preprocessing itself.

**Requested Changes:**

1. Narrow the theoretical claim, or substantially strengthen it. As written in the paper, Proposition 3.2 only compares against a restricted canonical HGNN family with relation-wise sum/mean aggregation, so the paper should either soften the “strictly more expressive” claim or prove a stronger result against richer HGNN classes.
2. Include absolute wall-clock training time, memory usage, and standard deviations for the main comparison tables.
3. Improve the "relation collapse" reasoning, like explaining why in some datasets it behaves differently (e.g., DBLP and IMDB are won by baselines, and AM is only marginally better than RGCN).
4. Clarify the sensitivity to encoding dimension (Fig. 12 shows the plot, but it needs to be cited and explained better) and random seeds (Table 6 has some results, but the analysis or explanation is missing) and the full hyperparameter selection protocol for all baselines in the main paper or appendix.

---

> ### Author Response · Authors · 2026-04-23
> **Reponse to Reviewer cZ7a (1/2)**
>
> We sincerely thank the reviewer for the careful reading of our manuscript and for the constructive and detailed comments. We are encouraged that the reviewer recognized the relevance of our research. We also appreciate the reviewer’s critical feedback on the scope of the theoretical claim and the experimental reporting. These comments helped us identify several places where the manuscript should be made more precise. Below, we respond to each requested change and describe the corresponding revisions.
>
> ## Response to Requested Change 1: Narrow the theoretical expressiveness claim
>
> We thank the reviewer for pointing out this important issue. We agree that the original wording could be interpreted as making a broader theoretical claim than what Proposition 3.2 formally establishes.
>
> To avoid overstating the result, we have revised the manuscript to make the scope of the theoretical claim much more precise (see the red-highlighted revisions in the Introduction and the theoretical discussion for representative examples). In particular, we now state that BG-HGNN is theoretically more expressive than the **canonical relation-aggregating HGNN class** considered in Proposition 3.2, while stronger architectures such as HGT or SlotGAT are included as **empirical baselines** to evaluate practical performance and efficiency rather than as models covered by the formal theorem.
>
> We are grateful for this comment, as it helped us present the theoretical contribution in a more accurate and carefully scoped manner.
>
> ## Response to Requested Change 2: Add absolute wall-clock time, memory usage, and standard deviations
>
> We thank the reviewer for this helpful suggestion. In the revised manuscript, we have expanded the experimental reporting, as requested, by updating Table 1 to include:
>
> - absolute wall-clock training time,
> - memory usage,
> - and standard deviations.
>
> The newly reported wall-clock time and memory results are consistent with our earlier findings and further support the efficiency advantage of BG-HGNN. In addition, the reported standard deviations are small, indicating that the performance comparisons are robust across runs.
>
> We are grateful to the reviewer for encouraging a more complete and reproducible experimental presentation.
>
> ## Response to Requested Change 3: Improve relation collapse reasoning and explain dataset-dependent behavior
>
> We sincerely thank the reviewer for this insightful comment.
>
> In our work, **relation collapse** refers to the phenomenon where canonical HGNNs lose the ability to distinguish between different relation types after cross-relation aggregation. This occurs when relation-specific messages, after transformation, become indistinguishable once combined through aggregation operators.
>
> We would like to clarify that, similar to other expressiveness limitations, **relation collapse is not guaranteed to occur, nor is it equally severe across all datasets**. Its practical impact depends on dataset characteristics such as the number of relations.
>
> For example, DBLP and IMDB are relatively simple heterogeneous graphs with only 6 relation types each. In such settings, relation-specific parameterization is less likely to suffer noticeably from relation collapse, and canonical HGNN baselines can therefore remain competitive or even outperform our method in terms of accuracy. This is consistent with the motivation and positioning of BG-HGNN: our method is particularly designed for **complex heterogeneous graphs with many relation types**, where both parameter explosion and relation collapse are more likely to become significant. This trend is also empirically supported by **Figure 3**, which shows that the advantage of BG-HGNN becomes more pronounced as the number of relations increases. On AM, BG-HGNN improves accuracy over RGCN only marginally, but it does so with substantially fewer parameters, indicating a significantly favorable efficiency--accuracy trade-off.
>
> We thank the reviewer again for this helpful comment. In the revised manuscript, we have incorporated this discussion into the empirical section to provide a clearer and more nuanced interpretation of the dataset-dependent results (see the red-highlighted revisions in the empirical study section in page 10).

---

> > ### Author Response · Authors · 2026-04-23
> > **Reponse to Reviewer cZ7a (2/2)**
> >
> > ## Response to Requested Change 4: Clarify encoding-dimension sensitivity, random seeds, and hyperparameter protocol
> >
> > We thank the reviewer for pointing out that the robustness results required clearer explanation, and we apologize for this oversight. Figure 12 and Table 6 were added during the revision process to address Reviewer RRux’s question regarding the behavior of the random dense encoding, but we did not clearly reference them in the main text. In the revised manuscript, we have expanded the discussion of the random dense encoding mechanism (see the red-highlighted revisions on Page 30) and added explicit references to these results in the main text (see the red-highlighted revisions on Page 11). The results in Figure 12 and Table 6 show that the random dense encoding mechanism used in our method is robust both to sampling randomness and to the choice of encoding dimension.
> >
> > In addition, we have added a clearer description of the hyperparameter selection protocol for BG-HGNN and all baselines in Appendix D.4. The appendix now includes a detailed description for each hyper-parameter such as hidden dimension, rank, encoding dimension, and model-specific parameters such as attention heads or slot numbers where applicable. We also clarify that all hyperparameters are selected based on validation performance under a comparable tuning budget. Furthermore, the code implementation is available through the anonymous link provided in the abstract and will be made publicly available upon acceptance.
> >
> > We sincerely appreciate this helpful suggestion, which allowed us to improve the experimental section.
> >
> > ## Closing
> >
> > We sincerely thank the reviewer again for the thoughtful and constructive feedback. The comments helped us make the theoretical claims more precise, strengthen the experimental reporting, and clarify the robustness and scope of the proposed method. We believe the revised manuscript is significantly clearer and more balanced as a result.

---

### Decision · Action_Editor_RdJK · 2026-05-12

**Recommendation:** Accept as is

**Audience:**

Yes

**Audience Explanation:**

Efficient learning on heterogeneous graphs is a critical challenge in graph representation learning, particularly as real-world graph datasets have significantly high relational complexity. Researchers, specifically those focused on GNN scalability, efficiency, and structured data, will find value in the ideas introduced here.

**Claims And Evidence:**

Yes

**Claims Explanation:**

One of the key strengths of the work is its rigorous evaluation of the proposed method BG-HGNN across 11 benchmark datasets, demonstrating significant improvements in parameter efficiency and training throughput, which is impressing because it is without sacrificing predictive accuracy.

The revision effectively addressed initial reviewer concerns regarding the scope of the theoretical claims.

In particular, by narrowing the "strictly more expressive" claim to apply specifically to the canonical relation-aggregating HGNN class (Proposition 3.2), the theoretical results are in alignment with empirical evidence. Furthermore, the addition of absolute wall-clock time, memory usage, and standard deviations makes it more transparent and convincing. The conceptual clarification of  relation collapse" supported by the a figure and low-rank interaction fusion ablation studies, confirms that the proposed architectural choices are both well-motivated and technically sound.